



# Emission characteristics of refractory black carbon aerosols from fresh biomass burning: a perspective from laboratory experiments

Xiaole Pan[1], Yugo Kanaya[2], Fumikazu Taketani[2], Takuma Miyakawa[2], Satoshi Inomata[3], Yuichi Komazaki[2],

Hiroshi Tanimoto[3], Zhe Wang[1,4], Itsushi Uno[4], Zifa Wang[1]

[1]State Key Laboratory of Atmospheric Boundary Layer Physics and Atmospheric Chemistry, Institute of Atmospheric Physics, Chinese Academy of Sciences, Beijing, 100029, China

[2]Japan Agency for Marine-earth Science and Technology, Yokohama, 236-0001, Japan

[3]National Institute for Environmental Studies, Tsukuba, 305-8506, Japan

[4]Research Institute for Applied Mechanics, Kyushu University, Kasuga, 816-8580, Japan

*Correspondence to*: Xiaole PAN (panxiaole@mail.iap.ac.cn)

**Abstract.** The emission characteristics of refractory black carbon (rBC) from biomass burning are essential information for numerical simulations of regional pollution and climate effects. We conducted combustion experiments in the laboratory to investigate the emission ratio and mixing state of rBC from the open burning of wheat straw and rape plants, which are the

main crops cultivated in the Yangtze River Delta region of China. A single particle soot photometer (SP2) was adopted to measure rBC-containing particles at high temporal resolution and with high accuracy. The combustion state of each burning case was indicated by the modified combustion efficiency (MCE), which is calculated using the integrated enhancement of carbon dioxide and carbon monoxide concentrations relative to their background values. The mass size distribution of the rBC particles showed a perfect Gaussian shape with an average mass equivalent diameter (MED) of 189 nm (the measured

MED values varied between 152 nm and 215 nm), assuming an rBC density of 1.8 g/cm$^3$. rBC particles less than 80 nm in size (the lower detection limit of the SP2) accounted for only ~5% of the total rBC mass, on average. The emission ratios, which are expressed as $\Delta$rBC/$\Delta$CO ($\Delta$ indicates the difference between the observed and background values), displayed a significant positive correlation with the MCE values and varied between 1.8 – 34 ng/m$^3$/ppbv. Multi-peak fitting analysis of the delay time ($\Delta t$, or the time of occurrence of the scattering peak minus that of the incandescence peak) distribution

showed that rBC-containing particles with rBC MED = 200 ± 10 nm displayed two peaks at $\Delta t$ = 1.7 μs and $\Delta t$ = 3.2 μs, which could be attributed to the contributions from both flaming and smoldering combustion in each burning case. Both the $\Delta t$ values and the shell/core ratios of the rBC-containing particles clearly increased as the MCE decreased from 0.98 (smoldering-dominant combustion) to 0.86 (flaming-dominant combustion), implying the great importance of the rapid condensation of semi-volatile organics. This study highlights that open biomass burning produces the majority of coated

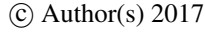



rBC particles, which have considerable ability to affect cloud processes and influence regional climate.

## 1 Introduction

Black carbon aerosols in the atmosphere play a vital role in the climate change by absorbing solar radiation and altering the formation and lifespan of clouds (Novakov et al., 2005;Ramanathan and Carmichael, 2008;Bond et al., 2013). It was

operationally defined according to its their light absorption capacity, chemical reactivity and/or thermal stability (Lack et al., 2014). One definition is refractory black carbon (rBC), which corresponds to the carbon mass derived from laser-induced incandescence (LII) emission at a boiling point at 4000 K. Open biomass burning (OBB) is one of the important sources of rBC, and it contributes ~42% of atmospheric loadings in the global emissions budget (Bond et al., 2004). In addition to rBC, OBB also simultaneously emits substantial amounts of semi-volatile organics (VOCs) that undergo extremely complicated

mixing processes with rBC in smoke during transport. Jacobson (2001) pointed out that the light absorption by internally coated rBC by inorganic/organic matter could increase, evidently due to the "lensing effect," in which a non-absorbing coating directs more light to the cores of rBC particles. However, the debate on the absorption enhancement capacity of rBC-containing particles is still ongoing, because discrepancies exist between observations and theoretical predictions based on Mie scattering models (Shiraiwa et al., 2010;Cappa et al., 2012) and among observation results from different locations

and using different sources (Healy et al., 2015;Liu et al., 2015b;Massoli et al., 2015;Ueda et al., 2016). The widely accepted explanations are as follows. First, the morphologies of rBC particles differ among different sources, and the process of particle aging in the atmosphere changes the physical structure of the particles. For instance, several studies have found that rBC particles with a fractal structure tend to collapse to a more closely packed shape when they are thickly coated (Adachi et al., 2010;Chen et al., 2010;He et al., 2015). Hygroscopic growth of rBC-containing particles also results in much more

compact rBC cores. Such a shrinkage effect is more likely to occur in polluted areas with high relative humidity (Fan et al., 2016). Second, the rBC particles are often located at off-center positions or may possibly be attached to the surfaces of non-rBC particles. Either of these circumstances renders the core-shell model invalid or introduces biases into the results. Sedlacek et al., (2012) reported that a large fraction (60%) of rBC-containing particles with non-core-shell structures exist in biomass burning plumes. Methodologies have also been developed to distinguish particles with attached rBC (bare rBC on

the surfaces of non-rBC particles) from rBC-containing particles with the core-shell structure (Moteki et al., 2014). In addition, OBB is an important source of brown carbon (BrC), which has distinct light absorbing features with different wavelength dependence, and their coexistence of rBC and BrC also influences the overall absorption enhancement of rBC-containing particles (Lack et al., 2012;Liu et al., 2015a;Saleh et al., 2015).

Biomass normally consists of celluloses/hemicellulose, organics and water. The emissions of rBC from OBB is determined

by the composition (carbon content) of the biofuel and the evolution of combustion (Yokelson et al., 1997;Andreae and Merlet, 2001). Briefly, the combustion process of biomass begins with the pyrolysis of biofuel molecules and the





evaporation of flammable mixtures (i.e., volatile compounds) and water, which is followed by flaming combustion that converts most carbon substances to carbon dioxide ($CO_2$). rBC particles are also produced in large quantities at this stage due to the oxygen-limited conditions and high temperatures. The last stage is smoldering combustion, which predominantly emits carbon monoxide (CO) and organics (Andreae and Merlet, 2001). Spherical organic particles without observable

nuclei (termed "tar-balls") are also reported to be associated with this smoldering combustion (Pósfai et al., 2004). Detailed descriptions of the physical properties of biomass burning particles have been provided in the literature (Reid et al., 2005;Akagi et al., 2011). The modified combustion efficiency (MCE), which is defined as $\Delta CO_2/(\Delta CO_2+\Delta CO)$ ($\Delta$ is the difference in measured concentrations between the OBB plume and the corresponding background value), has been used to indicate the relative amounts of flaming and smoldering combustion during a fire for characterizing the emission of rBC and

organic matter. An MCE value > 0.9 is normally regarded as flaming-dominant combustion, whereas MCE < 0.9 represents smoldering-dominant combustion (Kondo et al., 2011;Pan et al., 2012;Pan et al., 2013;May et al., 2014).

The rBC emission ratio ($\Delta rBC/\Delta CO$), which is defined as the enhancement of rBC with respect to its background versus that of CO, is an applicable indicator for quantifying the rBC emission intensity and constraining the rBC emission inventory for models (Pan et al., 2011). The observed values of $\Delta rBC/\Delta CO$ varied among different cases. For example, observations made

onboard the NOAA WP-3D aircraft yielded $\Delta rBC/\Delta CO$ values of $9 \pm 2$ ng/m$^3$/ppbv (Spackman et al., 2008)and 17.4 ng/m$^3$/ppbv (Schwarz et al., 2008) for fresh OBB plumes during the TexAQS field campaign. Airborne observations on the NASA DC-8 aircraft indicated that the $\Delta rBC/\Delta CO$ values were $8.5 \pm 5.4$ ng/m$^3$/ppbv and $2.3 \pm 2.2$ ng/m$^3$/ppbv for OBB plumes from Asia and North America, respectively (Kondo et al., 2011). Observations using a multi-angle absorption photometer (MAAP, which employs the filter-based light absorption technique; here we consider BC instead of rBC) at

mountain sites (30.16°N, 118.26°E, 1840 m a.s.l.) in South China yielded high $\Delta BC/\Delta CO$ values (10 - 14 ng/m$^3$/ppbv) when the site was subjected to OBB plumes (Pan et al., 2011). In fact, rBC emissions are heavily dependent on the combustion state of biomass. Field measurements using a semi-continuous ECOC analyzer (Thermo-Optical-Transmittance technique, IMPROVE protocol) during OBB episodes in East China yielded $\Delta EC/\Delta CO$ values of $17.4 \pm 5.2$ ng/m$^3$/ppbv for flaming-dominant cases and $11.8 \pm 2.3$ ng/m$^3$/ppbv for smoldering-dominant cases (Pan et al., 2012). Open burning experiments in

the laboratory on 15 individual plant species sampled in the United States indicated that $\Delta rBC/\Delta CO$ increased by up to 40 ng/m$^3$/ppbv as MCE increased to 0.95, and this result was largely insensitive to the biomass type used (May et al., 2014). The variability in $\Delta rBC/\Delta CO$ among observational studies also result from differences in sampling locations and conditions. For instance, airborne and ground-based measurements may capture different smoke parcels because flaming-dominant plumes are more easily injected to high altitudes than those released during the smoldering phase (Kondo et al., 2011). In

addition, $\Delta rBC/\Delta CO$ values decreased significantly with aging of OBB plumes, owing mostly to the coexistence with large fractions of water-soluble organic species (Mazzoleni et al., 2007;Gilardoni et al., 2016). Observations made in both East Asia and North America indicate a strong dependence of $\Delta rBC/\Delta CO$ on accumulated precipitation along backward trajectories (Kondo et al., 2011), implying that the rBC-containing particles became hydrophilic and were removed by wet





deposition during transport. This result is consistent with observations of OBB plumes at the top of Mt. Tai (36.26°N, 117.11°E, 1534 m a.s.l.) in North China, which indicated that ΔBC/ΔCO values for OBB plumes decreased substantially due to cloud scavenging processes (Pan et al., 2013).

The Yangtze River Delta Region (YRDR) is one of the most important agricultural regions in China, and it accounts for 29%
5 of total grain production. Wheat and rape plants are two major crops. After harvest, some of the crop straws are burned in the open air at fields, resulting in severe air pollution over the regional scale. Although field measurements on variations in the concentrations of rBC and CO and their ratios in OBB plumes have been reported, their physical characteristics may change significantly due to the rapid aging/mixing processes of semi-volatile organic vapors. Therefore, laboratory studies are very important to obtain insight into the initial emission features of rBC particles in smoke in China. In the present study,
10 we conducted open burning experiments using two crop residues (wheat straws and rape plants) obtained from the YRDR. The mass concentration and size distribution of rBC particles in the OBB plume were measured using a single particle soot photometer (SP2). The physical properties of nascent rBC-containing particles, the evolution of the size distribution, the mixing state of the rBC particles, and their dependence on the combustion state was investigated. The information presented in this paper is helpful for constraining/reducing uncertainties in OBB emission inventories and the estimates of their
15 climatic effects by models.

## 2 Experiments

### 2.1 Description of the burning experiments

We conducted burning experiments in the laboratory using samples of wheat straw and rape plants that were collected in an agricultural area of East China during a field campaign in 2010 (Pan et al., 2012). All of the biomass was stored in sealed
20 plastic bags to preserve its original state. A total of twenty-four samples (the mass of each sample was generally ~20 g) were ignited and burned on an aluminum foil net rack in a heat-resistant combustion box with an approximate volume of 144 L. The OBB smoke was removed from the room by a venting fan through a flexible rubber hose at a flow rate of 120 m$^3$/h. During this experiment, the wheat straw samples were classified into two groups. Twenty samples were burned in the chamber without artificial treatments, and four of the samples were placed in humid conditions (RH>99%) for 30 minutes to
25 absorb moisture for comparison. Detailed information on the setup of the OBB experiments is provided in the literature (Inomata et al., 2015). To monitor the evolution of the combustion process of biomass, the mixing ratios of $CO_2$ and CO in the OBB smoke were measured simultaneously. The mixing ratio of $CO_2$ was measured using a Li-7000 $CO_2$ analyzer (Li-COR Inc.; detection range 0–3000 ppmv, RMS noise 35 ppbv, integration time 0.5 s) through a separate 1/8 inch, ~1.5-m-long Teflon tube. The mixing ratio of CO was concurrently measured with an ultrafast CO analyzer (model AL5002, Aero-
30 Laser GmbH; detection range 0–100 ppmv, detection limit 1.5 ppbv, integration time 1 s). Comparison with a non-dispersive infrared CO gas analyzer (Thermo 48C; precision 10.0 ppbv, RMS noise 5.0 ppbv, average time 1 min) indicated that the measurement uncertainty was within 5%.





To avoid instrument overloading due to the extremely high concentrations of particles and trace gases, the inlets of the sampling lines were situated ~40 cm away from the combustion smoke. The sampling flows were subsequently diluted in a dilution system by mixing dry zero air produced by a zero gas generator (Thermo Inc., model 111). The flow rate of injected zero air was precisely controlled with a mass flow controller (Kofloc Inc., model 3660; accuracy ± 1.0% at 25°C). The

5 uncertainty of the dilution system was evaluated using known-size dry polystyrene sphere latex particles (PSL, Size Standard Particles, JSR Corporation, Japan). The PSL aerosols were produced by a nebulizer at a flow rate of 3.5 L/min and passed through a diffusion dryer (model 3062, TSI Inc., USA) and then size-selected using a differential mobility analyzer (DMA, model 3081, TSI Inc., USA). The number concentration of PSL particles was measured using a laser aerosol spectrometer (LAS-X II, PMS(GB) Ltd., UK; uncertainty 5%, flow rate 50 ccm) with and without dilution. The errors of the

10 dilution system were found to be 6%, 2%, 2%, 4%, and 5% for particles with mobility diameters of 120 nm, 200 nm, 300 nm, 500 nm, and 1000 nm, respectively, at a dilution ratio of ~50. During the burning experiment, a 50-cm-long, 1/4-inch flexible conductive silicone tube (TSI Inc., USA) and stainless steel Swagelok fittings were used for the aerosol tubing (dilution factor: 46), and polytetrafluoroethylene (PTFE) tube and fittings were used for the measurement of gases (dilution factor: 22). Abbreviations and symbols used in this paper is shown in Table 1.

**2.2 Instruments**

2.2.1 Single particle soot photometer

A single-particle soot photometer (SP2, Droplet Measurement Technologies Inc.) was used to examine the evolution of the number concentration and mixing state of the OBB particles. The SP2 employs a continuous intra-cavity Nd:YAG laser beam (1064 nm, TEM00 mode, Gaussian) to produce a strong laser power field and detects the laser-induced incandescence

signal emitted from individual rBC particles when they are heated to their boiling point (~4300 K) (Gao et al., 2007). The peak value of the incandescent signal was converted to the rBC mass based on a calibration curve determined using fullerene soot (FS) particles (stock 40971, lot: L20W054, Alfa Aesar, USA). The calibration procedure used in quantifying rBC masses was the same as that used in previous studies (Moteki and Kondo, 2010;Miyakawa et al., 2016). The effective density function of FS was determined on the basis of a DMA–aerosol particle mass (DMA-APM) system in Yutaka

Kondo's laboratory at the University of Tokyo and was consistent with previous results (Moteki and Kondo, 2007;Gysel et al., 2011) (Fig. S1). In our study, the mass of the individual FS particles ranged from 0.35 to 89.5 fg, which corresponds to 80–700 nm in mass equivalent diameter (MED), assuming an rBC density of 1.8 g/cm$^3$ and ideally spherical particles. Due to variations in morphology and composition under ambient conditions, the incandescent signal may not always be linearly proportional to the mass of rBC particles. The uncertainty of the derived rBC mass on the basis of the incandescent signal

was estimated to be ~ 30%, and the uncertainty of the derived MED values was estimated to be ~ 10%. The mass size distribution of the rBC in the OBB plume typically peaked at 180 ~ 200 nm (<10 fg). Extrapolation using a Gaussian curve fitted to the observed size distribution suggested that the missing rBC particles with MED < 80 nm and MED > 500 nm only cause minor mass underestimation (~5%). The size-dependent detection efficiency of SP2 for FS particles was also





evaluated on the basis of a DMA-SP2-CPC (model 3010, TSI Inc., USA) system, and we found that the SP2 detection efficiency was in the range of 0.94 ~ 0.98 for particles with MED values larger than 80 nm (shown in Fig. S2).

The scattering signal of the SP2 was calibrated using PSL particles with known sizes (170 nm, 200 nm, 254 nm, 300 nm, 500 nm and 1000 nm). It is worth noting that only PSL particles with sizes larger than 166 nm can be detected adequately,
implying that the ambient measurement of the SP2 will underestimate the total number concentration of light-scattering particles because the particles that are smaller than this size threshold are not counted. A two-elemental avalanche photodetector was employed in the SP2 to determine the actual position of the particle in the laser beam, which allows for delay time and coating thickness analysis of rBC particles with core-shell structure (Gao et al., 2007). Detailed information on the SP2 is provided in the literature (Schwarz et al., 2008;Schwarz et al., 2010;Moteki et al., 2014).

2.2.2 Determination of the coating thicknesses of rBC-containing particles

As mentioned above, when the rBC-containing particles pass through the laser beam, the rBC component needs a short period of time to absorb energy to gradually evaporate the coating materials. This means that the time when the rBC reached its boiling point was later than the time of occurrence of the peak of scattering. For the rBC-containing particles with core-shell structure, the coating thickness can be semi-qualitatively represented by the delay time of the LII peak ($\Delta t$), which is
defined as the elapsed time between the occurrence of the peak of the scattering signal and the peak of the incandescence signal; a positive value of this quantity indicates that the peak of the incandescence signal occurs after the peak of the scattering signal. In principle, the larger the $\Delta t$ value is, the more likely the rBC core is to be thickly coated. Such phenomena have been frequently reported in a number of studies (Moteki et al., 2007;Subramanian et al., 2010). In our experiment, a histogram analysis of the $\Delta t$ value demonstrated that there was a small $\Delta t^*$ peak at $0.8 \pm 0.5$ ($2\sigma$) μs for the
uncoated FS particle with MED < 400 nm. Apparently, this $\Delta t^*$ peak did not result from the coating effect, but the intrinsic error of the photodetector of the SP2. We regarded the particles with $\Delta t$ values larger than 1.3 (mean + $2\sigma$) μs as being coated. A detailed classification of thinly coated and thickly coated particles is discussed in section 3.4. In practice, estimation of the coating thickness of rBC cores in terms of $\Delta t$ values is sometimes problematic because, first, the $\Delta t$ values showed a discontinuous increase with an increase in coating thickness, depending on both the rBC core and the coating
material. A laboratory study of graphite particles coated with organic liquids indicated that the $\Delta t$ value jumped from less than 1 μs to 3 μs after the coating thickness of particles exceeded a threshold value (Moteki and Kondo, 2007). Such variations provide only a measure of the minimum detectable coating thickness, and they do not permit precise estimation. Second, the $\Delta t$ value can be negative in cases where the peak of the incandescence signal occurs *before* the peak of the scattering signal (Sedlacek et al., 2012). Negative $\Delta t$ values have been reported not only in biomass burning plumes
(Sedlacek et al., 2015) but also in laboratory experiments (Moteki and Kondo, 2007). An accepted explanation is that the rBC component is located at or near the surface of non-refractory matter (such particles are said to belong to the non-core-shell type or the attached type). When such particles passed through the laser beam, the rBC did not absorb a sufficient



amount of energy to evaporate the non-rBC substances, and the occurrence of fragmentation allowed some of the remaining non-rBC substances to pass though the laser, producing a scattering signal after the incandescence signal was induced. Moteki et al., (2014) suggested the use of the time-dependent variation of the scattering cross section ($C_s$) to discriminate the coated rBC particles from the attached type and proposed a measurable parameter, a logarithm of the ratio of $C_s$ before

evaporation ($C_{s-be}$) to $C_s$ at the onset of incandescence ($C_{s-oi}$), $\log(C_{s-be}/C_{s-oi})$, to quantify the contributions from the different types.

Following the same principle, we calculated the coating thickness of rBC-containing particles with the core-shell structure on the basis of the leading-edge-only (LEO) fitting method (Gao et al., 2007). The physical interpretation of this method has been described in the literature (Gao et al., 2007;Laborde et al., 2012). In the LEO fitting approach, the laser intensity profile

of the SP2 was predetermined from an analysis of the PSL particles, and the "leading edge" data are selected according to the criterion of $t < -2.5\sigma$. Here, $\sigma$ denotes the standard deviation of the Gaussian function of the laser intensity profile. Using a strict threshold (e.g., $t < -3\sigma$) can reduce the risk of the onset of evaporation of the coating; however, it significantly increases the possibility of incorrect Gaussian fitting. The optical diameter of the undisturbed rBC-containing particles was estimated using Mie scattering theory and the LEO-fitted scattering peak height, presuming a refractive index of $m = 1.5 –$

$0i$ for the coating materials. To test the validity of the LEO fitting method, laboratory experiments were performed using FS particles (Alfa Aesar 40971, lot: L20W054) coated with oleic acid (molecular weight 282.46, boiling point 360°C). The instruments used in this procedure are the same as those described in the literature (Moteki and Kondo, 2007). FS particles with mobility diameters of 100 nm, 150 nm, and 200 nm were selected using the DMA and then coated with oleic acid using a heated oleic oil bath. The coated particles with shell mobility diameters of 180 nm, 200 nm, 250 nm, 300 nm, and 350 nm

were selected by a second DMA and then measured using the SP2. We excluded the doubly-charged particles from the data analysis because both the core and shell sizes of these particles were apparently larger than those of the singly charged ones. The shell diameters of the coated particles were well determined using the LEO fitting method. Linear regression analysis of the calculated and the measured shell diameters demonstrated a good positive correlation with a high correlation coefficient ($r^2 = 0.9$), as shown in Fig. S3. The coating thickness of rBC was calculated using $(D_p − D_c)/2$, where $D_p$ and $D_c$ are the shell

and core diameters, respectively, of the rBC-containing particles. The shell/core (S/C) ratios were calculated using $D_p/D_c$. The total uncertainty of the S/C ratio values was calculated to be 14%.

### 2.2.3 Non-methane volatile organic compounds (NMVOCs)

In the present study, the mixing ratios of the NMVOCs in the gas phase in the OBB smoke were measured simultaneously using a high-sensitivity proton-transfer-reaction mass spectrometer (PTR-QMS 500, IONICON Analytik GmbH) with a time

resolution of 1.9 s. Four primary ions ($H_3^{18}O^+$, $NO^+$, $O_2^+$ and $H^+\cdot(H_2O)_2$) were adopted, and more than 20 product ions (NMVOC•$H^+$) were selectively monitored, according to predetermined multiple ion detection (MID) settings. The detection limit and dwell time of PTR-QMS for NMVOCs were generally less than 0.3 ppbv and 0.1 s, which are fully satisfactory for




the measurement of OBB smoke. Detailed information on the configuration of the instrument is provided in the literature (Inomata et al., 2015).

### 2.3 Determination of combustion states and the rBC emission ratios

As mentioned, the excess mixing ratios of CO ($\Delta CO$) and $CO_2$ ($\Delta CO_2$) are normally used to estimate the modified combustion efficiency (MCE) of biomass, which was defined as $\Delta CO_2/(\Delta CO_2 + \Delta CO))$. In the present study, the mixing ratios of CO and $CO_2$ were measured at a time resolution of 1 s. Although the real-time variation in MCE was obtained, comparisons were difficult because the combustion states and durations varied significantly among the different cases. Here, a fire-integrated modified combustion efficiency was calculated on the basis of equation (1).

$$MCE = \frac{\sum \Delta CO_2}{\sum \Delta CO + \sum \Delta CO_2} = \frac{\sum([CO_2]_{plume} - [CO_2]_{baseline})}{\sum([CO]_{plume} - [CO]_{baseline}) + \sum([CO_2]_{plume} - [CO_2]_{baseline})} \qquad (1)$$

where $\sum$ represents the time-integrated total mixing ratios of $\Delta CO$ and $\Delta CO_2$, and [X] is the mixing ratio of species X, expressed as ppmv. Correspondingly, the emission characteristics of rBC were indicated by the rBC emission factor $\Delta rBC/\Delta CO$, which was calculated on the basis of equation (2).

$$\Delta rBC/\Delta CO = \frac{\sum([rBC]_{plume} - [rBC]_{baseline})}{\sum([CO]_{plume} - [CO]_{baseline})} \qquad (2)$$

where [rBC] is the mass concentration of rBC in unit of $ng/m^3$. The baseline of the mass concentration of rBC and the mixing ratios of CO and $CO_2$ were determined from the linear interpolation of the data before and after each combustion experiment.

### 3 Results and discussion

### 3.1 Evolution of the combustion process

Figure 1 shows two sample time series of the number concentrations of rBC and non-rBC particles and the mixing ratios of CO and $CO_2$ in the smoke for a wheat straw combustion case (Fig. 1a) and a rape plant combustion case (Fig. 1b). The evolution of the concentrations of particles and gases were similar, although the carbon content and combustion duration differed between the two biomass types. This result indicates that the combustion process was overwhelmingly important in

determining the variations in emission characteristics. As described in the literature (Andreae and Merlet, 2001;Reid et al., 2005), smoldering combustion normally occurs after flaming-dominant combustion ceases, resulting in two isolated peaks for the mixing ratios of CO and $CO_2$. However, we did not observe such a clear boundary between these two stages in our burning experiments. This result occurred primarily because the combustion durations were short (less than 200 s), and both flaming and smoldering combustion might occur simultaneously in different parts of the biomass. The mixing ratios of CO



displayed a broader tail than those of $CO_2$, implying that the relative importance of smoldering combustion increased at the end; this effect is particularly clear in Fig. 1b.

The temporal variations in the number concentration of rBC were highly correlated with those of $CO_2$. This phenomenon is in accordance with previous conclusions that the production of rBC particles is mostly related to flaming-dominant combustion processes (Pan et al., 2012). In the high-temperature and low-oxygen environment present in the inner part of flames, the evaporated unsaturated alkanes tend to pyrolyze to soot precursor particles (i.e., PAHs), which is followed by the aggregation and considerable growth of the rBC particles. The non-rBC particles appear to be mostly emitted under low-temperature burning conditions (Reid et al., 2005). Directly after the biomass was ignited, a small peak (yellow shading in Fig. 1) in the number concentration was observed, which could be attributed to the condensation of semi-volatile matter released during the dry distillation step before glowing combustion. As flaming temperature increased up to 800K, rBC particles were produced in significant quantities, and these particles provided substantial numbers of condensation nuclei for semi-volatile compounds. As a result, the number concentration of non-rBC particles decreased to almost zero (red shading in Fig. 1). When combustion shifted from the flaming-dominant to the smoldering-dominant stage during the second half of the burning period (blue shading in Fig. 1), the number concentration of non-rBC particles increased again. This observation can be explained by the secondary condensation growth of pyrolyzed compounds that occurred because the temperature was not high enough to cause their complete oxidation. Overall, these observations confirmed the results of previous studies that indicate that particle formation in OBB is essentially a nuclei-limited condensation process, and rBC and $CO_2$ were mostly produced by flaming combustion, whereas organic matter and CO were emitted from the smoldering combustion (Reid et al., 2005).

Table 2 summarizes the information on the sample types, the mixing ratios of CO and $CO_2$, the mass concentration of rBC particles and MCE for all burning experiments. Here, the fire-integrated MCE value was used to represent the overall combustion condition for each combustion case because the combustion process differed significantly. As shown, the averaged MCEs have no significant differences for the combustion of dry wheat straw (0.86 ~ 0.98), wet wheat straw (0.88 ~ 0.96), and dry rape plants (0.91 ~ 0.96). The average mass concentration of rBC after 45 times dilution ranged from 0.25 to 19.8 μg/m³, and the average mixing ratio of CO after 22 times dilution ranged from 95 to 5003 ppbv.

### 3.2 Variations in rBC size as a function of MCE

The masses of rBC particles as a function of mass equivalent diameter (MED) displayed a perfect Gaussian distribution for all burning cases. For better expression, each dM/dlogDp distribution was normalized so that its maximum value = 1 (Fig. 2a). We found that the mass mode diameter (MMD) for each combustion case clearly increased from 152 nm to 215 nm as the overall MCE value increased from 0.862 to 0.964. The correlation ($r^2$ = 0.59) was significant at the 95% confidence interval (Fig. 2b). This result indicates that flaming combustion tends to produce larger particles than smoldering





combustion, which is consistent with previous studies that have suggested that intense flaming combustion cuts off the efficient transport of oxygen into the interior flame zone, resulting in considerable formation of small rBC particles. Because the coagulation rate of particles is roughly proportional to the square of their concentration (Lee and Chen, 1984), the growth in the size of rBC particles was consequently rapid under high concentration conditions. In this study, these

5 consecutive processes were interrelated and could not be decoupled in the analysis. Nevertheless, this mechanism explained the outliers (contained within the dashed ellipse shown in Fig. 2b), in that the number concentration of rBC particles was not sufficient to support substantial growth. For smoldering combustion, the production of rBC precursors (i.e., PAHs) was less effective because of the low temperature.

As mentioned, the MMD of rBC particles in OBB plumes was determined by the combustion condition; however, its

variations during the evolution of the OBB combustion process has not been fully investigated, because the separation of the combustion stage for *in situ* measurement was difficult. A recent study (Taylor et al., 2014) reported that the MMD of rBC particles was 152 nm. This value is apparently smaller than the frequently presented values (180 ~ 200 nm), and the authors attributed the differences to nucleation scavenging processes during transport. Published studies on OBB plumes are mostly based on airborne SP2 measurements. Andreae and Merlet (2001) noted that airborne measurements tend to be biased

toward flaming combustion because the plumes formed during the flaming stage was more likely to be injected to higher altitudes than those formed during the smoldering stage (Kondo et al., 2011). Another explanation involved rapid coagulation processes, in which small rBC monomers might easily form agglomerates or clusters driven by organic coatings when the temperature of the plume decreased. This process likely occurs so quickly (on the time scale of seconds) that ambient measurements (OBB plume age > 1 hour) cannot detect this process.

**3.3 Emission ratio of rBC particles**

Fig. 3 shows the dependence of the $\Delta rBC/\Delta CO$ ratio on the MCE for all combustion cases. For comparison, the results from previous OBB experiments in the laboratory, as well as from field measurements and emission inventories, are also plotted on the same figure. As shown, the $\Delta rBC/\Delta CO$ ratio increased from 1.5 $ng/m^3$/ppbv to 34 $ng/m^3$/ppb as the MCE increased from 0.91 to 0.98. The results of fitting a power function were similar with those from previous studies (McMeeking et al.,

2009;May et al., 2014), even though different types of biomass were combusted. This result indicates that the MCE value is a key parameter for determining the rBC emission intensity from OBB, irrespective of the difference in the types of biomass used. In the present study, we tested two biomass conditions (dry and wet) for wheat straw. For combustion cases involving wet wheat straw, we found that the values of the $\Delta rBC/\Delta CO$ ratio were always less than 7.1 $ng/m^3$/ppbv as the MCE value increased up to 0.96, and these values are much smaller than that (25.3 $ng/m^3$/ppbv) corresponding to dry wheat straw. This

result implies that the wet biomass was unfavorable for the production of rBC particles. This phenomenon is consistent with the experimental results described by Chen et al., (2010) who reported an evident decrease in the emission factor of elemental carbon and a moderate increase in the emission factor of CO for burning of moist wildland biomass.





In this study, the average $\Delta rBC/\Delta CO$ ratio was 13.9 ± 10.1 ng/m$^3$/ppbv when the biomass underwent flaming-dominant combustion (MCE > 0.95), and this value is higher than the value reported from airborne measurements (8.5 ± 5.4 ng/m$^3$/ppbv) for the outflowing aged OBB plumes observed in Asia during ARCTAS-A (Kondo et al., 2011) and the value (10 ± 5 ng/m$^3$/ppbv) for agricultural fires in Kazakhstan during ARCPAC (Warneke et al., 2009). In these studies, the OBB

plumes were sampled after they had undergone a week of transport. In-cloud scavenging may take effect at these relatively low values, although precipitation had not occurred. The $\Delta rBC/\Delta CO$ ratios derived from the FLAME-I and II experiments (McMeeking et al., 2009) are smaller (8.3 ± 9.7 ng/m$^3$/ppbv as converted from emission factors, expressed as g species kg$^{-1}$ dry fuel), probably because the combustion was inclined to smoldering (average MCE = 0.92). *In situ* measurements of OBB plumes at the location where our samples for burning were collected (Pan et al., 2012) indicated that the $\Delta EC/\Delta CO$ ratio

from flaming-dominant burning was 17.4 ± 5.2 ng/m$^3$/ppbv. This result approximates those obtained in the laboratory. Measurements of OBB plumes in the North China Plain found that the $\Delta EC/\Delta CO$ ratios from wheat straw burning ranged from 15 to 17 ng/m$^3$/ppbv (Pan et al., 2013). Kondo *et al*. (2011) reported $\Delta rBC/\Delta CO$ ratios as low as 2.86 ± 0.35 ng/m$^3$/ppbv (MCE = 0.96) for a fresh OBB plume in North America. These large discrepancies might result from significant rBC losses during transport. The yellow shading in Figure 3 indicates the variability of $\Delta rBC/\Delta CO$ ratios adopted in

emission inventories. Comprehensive analyses including all kinds of OBB showed that the $\Delta rBC/\Delta CO$ ratios were 8.6 ± 1.2 ng/m$^3$/ppbv (as converted from emission factors, assuming a molar volume of 22.4 L at standard temperature and pressure conditions for CO) (Andreae and Merlet, 2001), 7.5 ± 1.3 ng/m$^3$/ppbv (Akagi et al., 2011), and 9.0 ± 1.6 ng/m$^3$/ppbv using the bottom-up method (Yan et al., 2006). Synoptically speaking, a relatively high rBC emission ratio was suggested for estimating emission inventories of OBB because the majority of rBC emissions normally occur during the flaming

combustion stage, despite the longer duration of the smoldering stage.

The emissions of rBC relative to $\Delta CO_2$ were also calculated for each burning case (see Table 2). In general, the $\Delta rBC/\Delta CO_2$ ratios did not show a clear increase with increasing MCE values, and they varied from 149 to 1541 ng/m$^3$/ppmv, with a mean value of 592.5 ± 364.1 ng/m$^3$/ppmv. Schwarz et al. (2008) reported a high $\Delta rBC/\Delta CO_2$ ratio of 1770 ± 400 ng/m$^3$/ppmv, much higher than the values obtained in this study. Small values were also reported for OBB plumes observed

in North America (100–357 ng/m$^3$/ppmv), Siberia (167 ng/m$^3$/ppmv) (Kondo et al., 2011), and China (245 ng/m$^3$/ppmv) (Pan et al., 2012). Despite the differences among these values, the $\Delta rBC/\Delta CO_2$ ratio remains a useful parameter in constraining the uncertainty of emissions of rBC from OBB to within an order of magnitude in models.

### 3.4 Delay time of incandescence

For the rBC-containing particles, the delay time ($\Delta t$) of the peak of the incandescent signal *after* that of the scattering signal

is widely accepted as a proxy for the coating thickness of rBC particles. Particular caution should be employed when this concept was applied in data exploration. First, SP2 cannot detect particles with shell diameters less than 166 nm (the lower detection limit of SP2) owing to their weak scattering signal. A systematic bias always occurs for investigating the coating of rBC-containing particles with small rBC cores (i.e., MED less than 100 nm) because only very thickly coated rBC





particles can be counted. To avoid this bias, we report only the rBC-containing particles with relatively large rBC cores (MED = 200 ± 10 nm). Second, only the rBC-containing particles with positive Δt values were technically deemed as having a core-shell structure, to which the delay time-based method and LEO fitting method could be appropriately applied. In general, the scattering profile of rBC particles in the core-shell structure contained a main peak with a shoulder peak, which

resulted from the coating material and the rBC core, respectively. If there was only one predominant scattering peak with a quasi-Gaussian shape, it suggested that the evaporation of non-rBC coatings was insignificant, and the rBC-containing particle likely belonged to the attached type, as demonstrated in the literature (Moteki et al., 2014). In this study, the shell diameters of rBC-containing particles with the core-shell structure were estimated using the LEO fitting method (described in section 2.2.2). We found that the shell diameters of rBC-containing particles with rBC cores having MED = 200 nm

ranged from 210 to 400 nm, with a $5^{th}$ percentile value of 218 nm and a $95^{th}$ percentile value of 330 nm. The corresponding shell/core (S/C) ratios were between 1.09 and 1.7.

The dependencies of Δt on the derived S/C ratios of all of the coated rBC particles for all of the burning experiments are shown in Fig. 4. The color in the plot represents the total number count of particles in each bin. In general, Δt increases as the S/C ratio increases, reflecting that the rBC particles must spend a longer period of time absorbing energy to evaporate

thicker coatings. Histogram analysis showed that both the S/C ratio and Δt displayed neither simple Gaussian nor lognormal distributions. Instead, a multiple-peak Gaussian provided a good fit to their number distributions (Fig. 4). For the distribution of the S/C ratio, there were two modes. One mode (#1) occurs at an S/C ratio = 1.18, and another mode (#2) occurs at an S/C ratio = 1.34, indicating that the rBC particles had different levels of coatings. The differences in coating thickness were most likely related to the combustion state of biomass burning. As mentioned, although flaming combustion

emitted a substantial amount of semi-volatile organic carbons, the production of rBC particles was also significant at high temperatures. The competing condensing processes under nuclei-rich conditions resulted in relatively thin coatings on the rBC particles, instead of direct formation of particulate organic matter. Previous studies (Kudo et al., 2014;Inomata et al., 2015) also found that the emission factor of NMVOCs during the flaming stage was lower by an order of magnitude than that during the smoldering stage. It also supported our conclusion that the thinly coated rBC particles associated with mode

#1 and mode #2 were primarily related to the flaming combustion and smoldering combustion stages, respectively. In the present study, the integrated area ratio between mode #1 and mode #2 was 0.76. This result suggested that the thinly and thickly coated rBC particles were almost the same. It was worth noting that the histogram of the delay times of rBC-containing particles also had two modes, with one mode occurring at Δt = 1.74 μs and another peak occurring at Δt = 3.18 μs. The integrated area ratio between these two modes was 0.78, almost the same as the ratio derived from the S/C mode. This

result demonstrates that the rBC-containing particles with rBC cores of 200 nm and S/C ratios = 1.18 and 1.34 corresponded to delay times of 1.74 μs and 3.18 μs, respectively, at least in this study. Moteki et al., (2007) investigated the relationship between delay time and coating thickness for ambient rBC particles with MED = 200 nm, and they reported that the delay time increased linearly from 0-1 μs to ~4 μs as the coating thickness increased up to 200 nm (S/C ratio = 2), consistent with



our study. It should be noted that the delay time for uncoated rBC particles (S/C = 1) was not necessarily zero because of intrinsic differences among SP2 instrumentations. For instance, a shift of $\Delta t^* = 0 \sim 0.6$ µs was found for uncoated rBC particles among studies (Moteki et al., 2007;Sedlacek et al., 2012). By subtracting the $\Delta t^*$ value (0.8 µs) in this study, the $\Delta t$ was found to be $0.9 \sim 2.4$ µs for coated rBC particles in the biomass burning plumes.

**3.5 Coating thickness of rBC as a function of MCE**

Fig. 5a depicts the variations in modal $\Delta t$ values for rBC particles with MED = 200 nm as a function of MCE value. As shown, the $\Delta t$ clearly decreased as the MCE value increased ($r^2 = 0.44$), and the S/C ratio decreased from 1.4 at MCE < 0.9 (smoldering combustion) to 1.2 at MCE > 0.95 (flaming combustion). The decreasing trend was statistically significant at a level of 5%. As a matter of fact, the variations in both the $\Delta t$ values and the S/C ratios as a function of MCE showed similar tendencies for particles with MED of the rBC cores ranging between 190 nm to 250 nm. Our results differed from those of airborne measurements during the ARCTAS campaign, which indicated a tendency for the values of the S/C ratio to increase with MCE (Kondo et al., 2011). It is difficult to propose an explanation for this phenomenon because the observation conditions and OBB source regions are quite different.

Statistically, the modal coating thickness of rBC particles was found to be ~20 nm (Fig. S4). In fact, discrepancies exist among studies due to differences in biomass types, burning conditions and sampling locations. For example, ambient measurements in Europe indicated a coating thickness of rBC particles of 15 nm on average, and more than half of the rBC particles had a coating thinner than 10 nm (Laborde et al., 2013). Airborne measurements reported a thicker coating ($65 \pm 12$ nm) for rBC particles from brush fires (Schwarz et al., 2008).

Table 3 summarizes recent studies that report the coating thicknesses and S/C ratios of rBC particles. We found that the aging of particles was more important than their sources in determining the coating thickness of rBC particles. The coating thickness of freshly emitted rBC particles from OBB was relatively small (~ 20 nm), and this thickness was reported to increase to $65 \pm 12$ nm (Schwarz et al, 2008) and up to 100 nm (Taylor et al., 2014) when they experienced transport over hours or days. The coating thickness of rBC particles from traffic emissions varied significantly, depending on the urban emission and photochemical processes. The transport pattern and meteorological parameters (such as RH) also play a role in changing the morphology of rBC-containing particles. Fan et al., (2016) reported that the hygroscopic shrinkage effect under high RH conditions led aggregated soot particles to become more tightly clustered, which resulted in an increase in the shell/core ratios. Such phenomena have also been observed under ambient conditions (Adachi et al., 2010). It has also been reported that condensation and coagulation processes cause the voids of rBC aggregates to be filled or cause the particles to collapse into a compact shell-core structure (Zhang et al., 2008).

**3.6 Relationship between the S/C ratio and NMVOCs**

Scatter plots of the S/C ratios of rBC particles with core MED = 200 nm plotted against the emission factor (EF, in unit of g/kg) of each condensable NMVOC are shown in Fig. 6. It is obvious that the S/C ratio increases with the EF for the burning



of both dry wheat straw and dry rape plants. As discussed in a previous study (Inomata et al., 2015), the high EF values of the NMVOCs are related to smoldering combustion, in which the production of rBC particles is less effective due to the low-temperature conditions. The condensation of semi-volatile organics was evident under rBC nucleus-limited conditions, resulting in higher S/C ratios. We also found that the S/C ratio from the open burning of rape plants was apparently higher

than that of wheat straw under the same NMVOC emissions conditions. This difference is likely attributable to the physical formation processes of rBC particles under high-temperature conditions for different combustion types. For instance, intensive fires tend to produce non-spherical rBC particles with chain-like structures, which display uneven coatings of semi-volatile organics on the rBC particles; however, for compact, quasi-spherical rBC particles, it is much easier to form evenly distributed, thicker coatings. We hypothesize that the open burning of rape plants tends to produce more compact

rBC particles. In addition, for wheat residues, the S/C ratios (~1.4) of rBC for wet samples was apparently higher than those (12~ 1.3) of dry material at the same EF. This result indicates that the microphysical properties of the rBC particles varied under different biomass burning conditions. To answer these questions, further analysis of individual particles using electro-microscopy is urgently needed.

**4 Conclusions**

In the present study, biomass burning experiments were conducted in the laboratory using wheat straw and rape plants, two major agriculture crop residues, which were obtained from the Yangtze River Delta Region (YRDR) during the Rudong field campaign. The combustion state of each biomass burning experiment was assessed using the modified combustion efficiency (MCE) that was calculated on the basis of fire-integrated excess CO and $CO_2$ mixing ratios, relative to their background values. A full calibration of the single particle soot photometer (SP2) was performed using commercial

calibrating particles (fullerene spheres and polystyrene latex particles) following the standard procedures that were proposed by Moteki and Kondo (2007). The mass equivalent diameter (MED) values of the rBC particles were calculated on the basis of the measured incandescence signals and a presumed rBC density of 1.8 $g/cm^3$. The major findings are as follows. (1) The emission ratio of rBC, defined as $\Delta rBC/\Delta CO$, increased from 0.2 $\mu g/m^3$/ppbv to 40 $\mu g/m^3$/ppbv as MCE increased from 0.84 to 0.98. The increasing trend was in accordance with previous laboratory studies, and this result was largely insensitive to

the type of biomass considered. (2) The mass mode diameter of the rBC particles ranged from 152 to 215 nm. It was found that the rBC particles produced during flaming combustion had obviously larger MMD values than those produced during smoldering combustion. This result indicated that the coagulation/growth process was intensive during flaming combustion as a result of substantial production of rBC precursor particles, such as PAHs, under the high-temperature and oxygen-deprived conditions that occurred within intense flames. (3) For rBC-containing particles, the delay time of the occurrence

of the incandescent peak *after* the scattering peak clearly increased as the S/C ratio increased, and smoldering combustion tends to produce more thickly coated rBC particles than flaming combustion. (4) The condensation of semi-volatile organics co-emitted by the thermal pyrolysis of biomass plays a key role in the formation of coatings on the surface of rBC particles,



though the mixing/coating processes of condensable NMVOCs may vary significantly, probably due to the distinct physical characteristics of rBC particles produced by the burning of wheat straw and rape plants.

**Acknowledgements**

This work was supported by the National Nature Science Foundation of China (Grant No. 41675128 and No. 41225019).

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



# Tables

Table 1 Abbreviations and symbols used in this paper.

| Symbol/acronym | Full name/explanation |
| --- | --- |
| rBC | Refractory black carbon, as derived using the LII method at a temperature of ~ 4000K. |
| $\Delta t$ | The delay in the time of occurrence of the incandescence peak *after* that of the peak of the scattering signal |
| APM | Aerosol particle mass analyzer (Kanomax Inc.) |
| C/S ratio | Shell/core ratio |
| CO | Carbon monoxide |
| $CO_2$ | Carbon dioxide |
| $C_S$ | Scattering cross-section |
| DMA | Differential mobility analyzer (TSI Inc.) |
| FS | Fullerene soot (C60) |
| LEO fitting | Leading-edge-only fitting method proposed by Gao et al., (2007) |
| LSP | Light scattering particle |
| MCE | Modified combustion efficiency |
| MED | Mass equivalent diameter |
| MMD | Mass mode diameter |
| non-rBC | Non-refractory black carbon matter that evaporates as rBC absorbs energy |
| OBB | Open biomass burning |
| SP2 | Single particle soot photometer (DMT Technologies) |



Table 2 Description of sample types; overall combustion states; CO, CO₂, and rBC concentrations; MED of rBC; and emission ratios for the burning experiments. The ordering of these quantities is the same as in previous studies (Inomata et al., 2015).

| Experiment. | Sample type | MCE | $CO^*$ | $CO_2$ | $rBC^{**}$ | $rBC/CO$ | $rBC/CO_2$ | MED of rBC |
|---|---|---|---|---|---|---|---|---|
| | | | ppbv | ppmv | $ng/m^3$ | $ng/m^3/ppbv$ | $ng/m^3/ppmv$ | nm |
| 1 | Wheat straw/dry | 0.964 | 1,181 | 695 | 13983 | 24.2 | 904.8 | 215 |
| 2 | Wheat straw/dry | 0.930 | 311 | 91 | 2366 | 15.6 | 1171.5 | 188 |
| 3 | Wheat straw/dry | 0.952 | 261 | 114 | 1446 | 11.3 | 570.4 | 152 |
| 4 | Wheat straw/dry | 0.949 | 343 | 141 | 4812 | 28.7 | 1541.0 | 187 |
| 7 | Wheat straw/dry | 0.953 | 256 | 114 | 1408 | 11.2 | 554.6 | 160 |
| 8 | Wheat straw/dry | 0.976 | 380 | 340 | 6290 | 33.9 | 832.6 | 191 |
| 9 | Wheat straw/dry | 0.917 | 189 | 46 | 168 | 1.8 | 165.3 | 187 |
| 10 | Wheat straw/dry | 0.944 | 274 | 101 | 787 | 5.9 | 348.9 | 148 |
| 11 | Wheat straw/dry | 0.862 | 464 | 64 | 470 | 2.1 | 331.8 | 152 |
| 12 | Wheat straw/dry | 0.937 | 230 | 75 | 429 | 3.8 | 256.5 | 148 |
| 13 | Wheat straw/dry | 0.950 | 282 | 118 | 1790 | 13.0 | 682.8 | 163 |
| 14 | Wheat straw/dry | 0.952 | 290 | 126 | 1295 | 9.1 | 461.1 | 160 |
| 15 | Wheat straw/wet | 0.909 | 4,757 | 1,045 | 11255 | 4.8 | 484.5 | 196 |
| 16 | Wheat straw/wet | 0.904 | 1,339 | 277 | 8658 | 13.2 | - | 177 |
| 17 | Wheat straw/wet | 0.961 | 75 | 41 | 250 | 6.8 | 276.2 | 148 |
| 18 | Wheat straw/wet | 0.884 | 1,373 | 230 | 5350 | 8.0 | 1045.7 | 181 |
| 19 | Rape plant/dry | 0.943 | 2,702 | 983 | 19802 | 15.0 | 906.0 | 204 |
| 20 | Rape plant/dry | 0.923 | 3,512 | 926 | 13402 | 7.8 | 651.2 | 189 |
| 21 | Rape plant/dry | 0.909 | 286 | 63 | 209 | 1.5 | 149.6 | 137 |
| 22 | Rape plant/dry | 0.951 | 956 | 408 | 3052 | 6.5 | 336.4 | 155 |
| 23 | Rape plant/dry | 0.960 | 457 | 241 | 1259 | 5.6 | 234.9 | 142 |
| 24 | Rape plant/dry | 0.954 | 846 | 386 | 4609 | 11.1 | 537.4 | 144 |

5   *: the mixing ratio of CO was diluted by 22 times.

**: the mass concentration of rBC was diluted by 45 times.



Table 3 Brief summary of the coating thickness and shell/core (S/C) ratios for rBC emissions from different sources collected from recent studies.

| rBC source | Coating thickness | S/C ratio | rBC core size | Age | Sampling description | Study |
|---|---|---|---|---|---|---|
| Biomass burning | 65 ± 12 nm | - | *190~210 nm | 0.5~1.5 hour | Airborne SP2 measurements during 2006 Texas Air Quality Study | (Schwarz et al., 2008) |
| | ~ 15 nm | | 200 nm | - | Field measurements using SP2 in the agglomeration of Paris as part of MEGAPOLI European project | (Laborde et al., 2013) |
| | 50 ~ 100 nm | 2.0 ~ 2.5 | 152~196 nm | 1~2 days | Airborne SP2 measurement during the second phase of the BORTAS project over Eastern Canada and the North Atlantic during July-August 2011. | (Taylor et al., 2014) |
| | - | 1.3 ~ 1.6 | **120 ~ 140 nm | 1~2 hours | Airborne SP2 measurements during ARCTAS in spring and summer | (Kondo et al., 2011) |
| | 20 nm | 1.2 ~ 1.4 | 200 ±10 nm | < 10 s | Burning experiments in combustion chamber in laboratory environment | This study |
| Asia continental | - | 1.6 | 200 nm | 2~3 days | Ground-based SP2 measurements at Fukue Island, Japan | (Shiraiwa et al., 2008) |
| Free troposphere | - | 1.3 ~ 1.4 | 200 nm | 12 hours | | |
| Aged air mass | 44 nm | - | 200 nm | | Field measurement using SP2 in the agglomeration of Paris as part of MEGAPOLI European project | (Laborde et al., 2013) |
| Traffic influence | 2 ±10 nm | | 200 nm | | | |
| Traffic emission | 110~300 nm | | 80~130 | Highly aged | Ground-based SP2 measurement at Urban site at Shanghai, China | (Gong et al., 2016) |
| Traffic emission | - | 1.6 ~ 2.4 | - | - | Clean Air for London (ClearfLo) experimental campaign in winter, 2012 | (Liu et al., 2014) |
| Solid fuel burning | - | <1.2 | - | - | | |
| Europe continental | - | 1.45 ~ 1.6 | - | - | | |
| Urban emission | 20 ~ 30 nm | | > 200 nm | 1~2 days | Airborne SP2 measurement during MILAGRO campaign | (Subramanian et al., 2010) |

*: Volume equivalent diameter; ** Count Median Diameter



# Figures

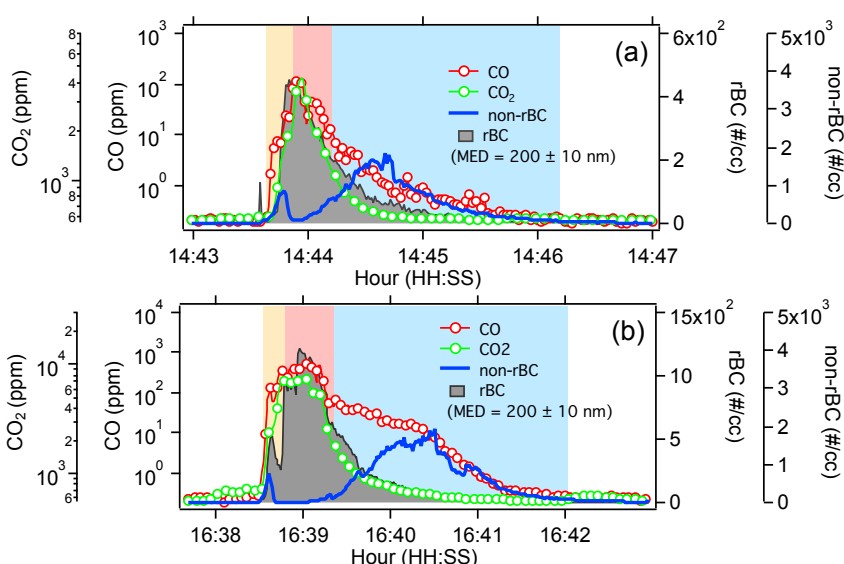

5    Figure 1: Temporal variations in the mixing ratios of CO, CO$_2$, the number concentrations of rBC and non-rBC particles for the burning of
wheat straw (a) and rape plants (b). The yellow (dry distillation step of the biomass), red (flaming-dominant combustion) and blue
(smoldering-dominant combustion) shaded areas in the plot represent the different burning states.




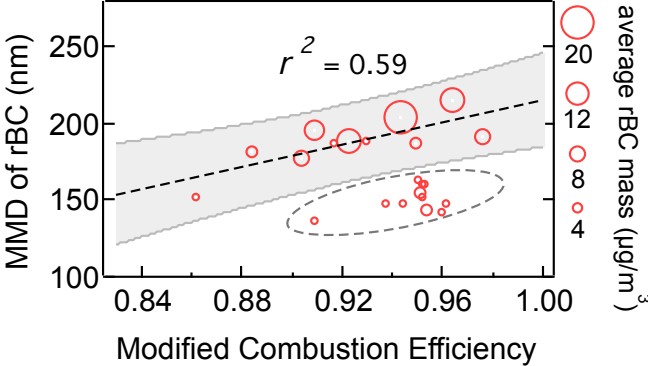

Figure 2. Variation of mode mass equivalent diameter (MMD) as a function of the modified combustion efficiency. The size of each circle indicates the average rBC mass concentration for each burning case.

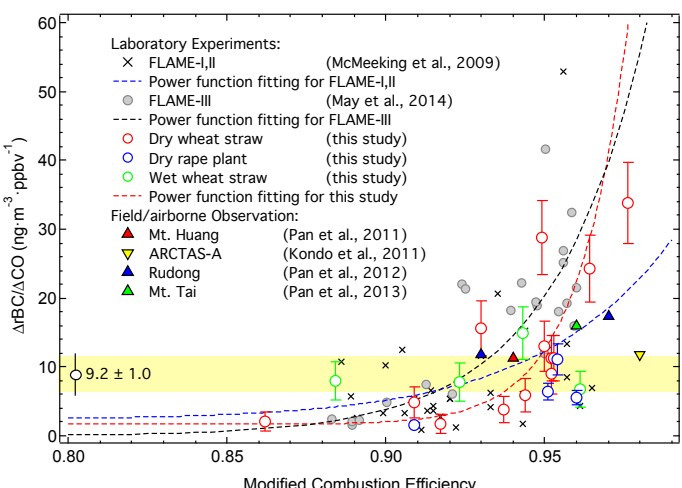

Figure 3. The variations in the emission ratio of rBC and ΔrBC/ΔCO, as a function of averaged MCE for all burning cases. Previous observations and the results of laboratory burning experiments are displayed in the plot.





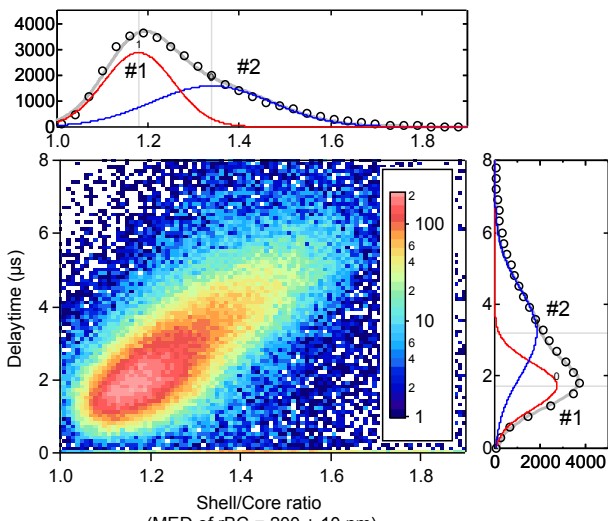

5  Figure 4. The dependence of delay time of the peak of incandescence signal *after* that of the scattering signal as a function of
the shell/core ratio for the rBC particles with MED = 200 ± 10 nm for all the burning cases, and the multiple Gaussian fitting
for all the data of cross-sections along the x-axis and y-axis.





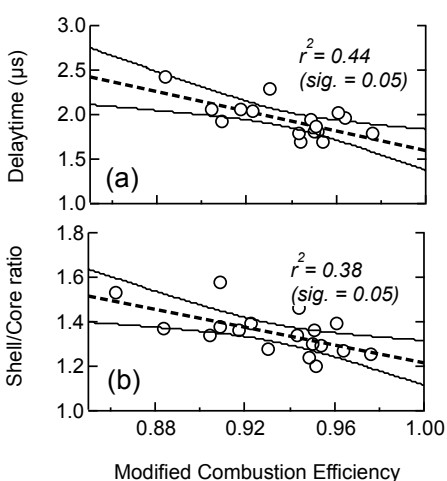

Figure 5. Variations in the delay time (a) and the shell/core ratio (b) as a function of MCE values for rBC particles with MED = 200 ± 10 nm.





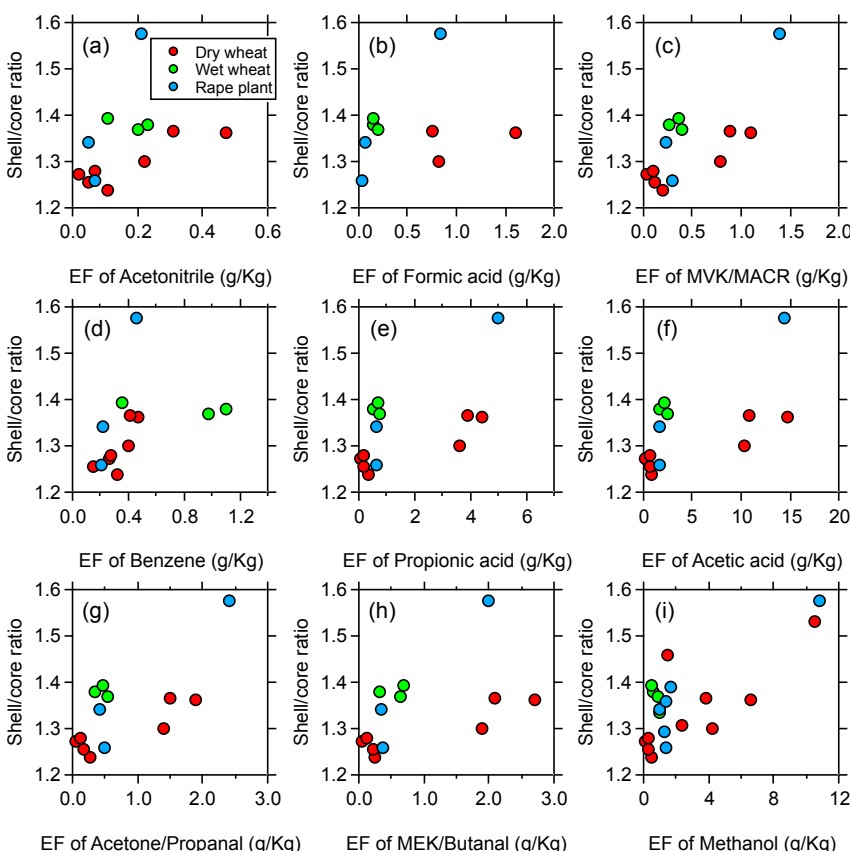

Figure 6. Variations in the shell/core ratios of rBC particles with MED = 200 ±10 nm as a function of the emission factor of each experiment. Here, EF is defined as the amount of each compound released per unit amount of dry fuel consumed. The
5   red, green and blue colors indicate the dry wheat straw, wet wheat straw and rape plant samples, respectively.