# Peer review of "Emission characteristics of refractory black carbon aerosols from fresh biomass burning: a perspective from laboratory experiments"

_Atmospheric Chemistry and Physics, 2017_

## Referee Comment (RC1) · Anonymous Referee #1 · 23 May 2017

Summary: The present manuscript describes a series of laboratory experiments on wheat straw and rape or rapeseed plants to quantify emission ratios and mixing state of refractory black carbon (rBC) using the Single Particle Soot Photometer (SP2). The ultimate objective of this study is to provide data that can augment field measurements of biomass burning (BB) events as well as providing important BB inventory used by models. The primarily findings of value to the community include the quantification of the delta-rBC/delta-CO ratio as a function of MCE (modified combustion efficiency); dependence of rBC mass mode diameter on MCE; and dependence of rBC mixing state on MCE for the two fuel sources studied. The biggest disappoint of this study is the lack of measurements of the rBC optical properties as a function of MCE. From a climate forcing point-of-view, quantification of BB optical properties is central to bounding the contribution of these events to aerosol radiative forcing. Consequently, this study is very myopic - only two fuels are examined and the experiments carried out are rBC-centric. This being said, the data is expected to find value in emission inventories and thus should be considered for publication after comments, listed below, are addressed.

The authors correctly indicate that the combustion process is the driving force that dictates variations in emission characteristics. It is clear that the authors characterized stages of the burn as either flaming or smoldering, yet offer no boundary conditions as to when a burn was flaming vs smoldering. If MCE was used, what value determined if the data points were from an active flaming condition or smoldering condition? And, if as the authors point out, both stages could occur at the same time. On page 9, line 13, the authors state that "when the combustion shifted from the flaming dominant to the smoldering-dominant state. . ." what is the criteria used to characterize one stage over the other?

This reviewer is rather surprised that the authors fit the rBC size/mass distribution data to a Gaussian (page 1, line 19; page 9, line 27) as opposed to a lognormal. It is well-known that most aerosol distributions are skewed (e.g., exhibit a long tail at larger sizes) and thus are better described with a lognormal function (Hinds 1999). On page 9, line 28, the authors make reference to figure 2a that presumably shows an example dM/dlogDp plot. Such a plot does not exist. The authors are strongly encouraged to add this figure along with a lognormal fit.

Central in their study is the use of MCE. The authors are encouraged to read the 2016 publication by Collier et al., (Regional Influence of Aerosol Emissions from Wildfires Driven by Combustion Efficiency: Insights from the BBOP Campaign; (2016) Environ. Sci. Technol.50, 8613−8622) with specific attention to Figure 4). While the Collier paper focuses on wildfires, the dependence of aerosol emissions on MCE the authors might might this study relevant to theirs.

Page 1, Line 15: The authors cite in their abstract that "A single particle soot pho-tometer (SP2) was adopted to measure rBC-containing particles at high temporal res-olution and with high accuracy," yet do not explicitly discuss what was "adopted" to realize the high temporal resolution and high accuracy. If the authors altered some hardward/software aspect of the SP2 that improved upon its "out-of-the-box" capabili-ties, then they should explicitly discuss those changes. If nothing was not done, than eliminate this statement as it is misleading.

Page 4, line 19: The authors write " All of the biomass was stored in sealed 20 plastic bags to preserve its original state." Sealing a sample will not prevent loss of semi-volatile materials. Have the authors accounted for this or attempted to quantify this?

Page 4, line 22: "flexible rubber hose". This reviewer assumes that the authors mean "conductive" tubing. If so, please state that.

Page 4, line 24/25: The authors indicate that four samples were placed in humid con-ditions for 30-minutes to absorb moisture. How was the moisture content quantified? Why only 30-minutes? What was the goal? To 'coat' the fuel with some moisture or increase the moisture content of the fuel? The moisture content would be expected to potentially impact the MCE and, in turn, the rBC/CO ratio and thus better quantification would be warranted.

Page 7, line 14/15: The authors assume that the non-refractory coating possesses a refractive index of 1.5 - 0i. While likely valid, the authors are encouraged to acknowl-edge that while BB events are a major source of brown carbon (BrC), it is highly unlikely that shortwave light absorbing OA will absorb at 1064 nm - the laser wavelength utilized by the SP2.

Page 9, line 28/29: As stated above, Figure 2a does not exist.

Page 10, line 25 - 28. While some trends appear to be present, the lack of water content quantification limits how much can be concluded with respect to comparing dry

and wet wheat straw. The authors are encouraged to address this either by estimating the change in water content that a 30-minute exposure of a 99% RH environment could create or acknowledge that the lack of water content quantification limits the quantitative comparison of dry and wet wheat straw emission ratios.

Page 11/12 and Figure 4. The linearity between the incandescence delay time and shell/core ratio is somewhat surprising. In the application and comparison of the two methods of analyzing the rBC mixing state - incandescence delay and coating thickness - was the LEO method applied to the incandescence delay time analysis? Not only will the LEO method impact the scattering signal amplitude, it could impact the scattering peak location relative to the rBC incandescence peak. Therefore, as the shell/core ratio increases evaporative losses might be expected to exert a greater impact the location of the uncorrected and LEO corrected delay times.

Page 13, line 19/20: The authors state that "The coating thickness of freshly emitted rBC particles from OBB was relatively small ($\sim$ 20 nm), and this thickness was reported to increase to $65 \pm 12$ nm (Schwarz et al, 2008) and up to 100 nm (Taylor et al., 2014) when they experienced transport over hours or days." The authors are cautioned here. As Schwarz et al. state: "Although the sources of these emissions are unknown, their location and season of occurrence suggest that neither BB plume is from agricultural sources, but from brush fires." Similarly, Taylor et al., interrogated a boreal forest fires. The source fuel examined by the authors is agricultural in origin, not wilfires. Therefore caution must be exercised when extrapolating to expected aging behavior using two very different source fuels. As a matter of fact, this Reviewer is not convinced of the statement "We found that the aging of particles was more important than their sources in determining the coating thickness of rBC particles." More discussion is needed to buttress this statement.

Please insert error bars on Figure 6 if possible

---

## Referee Comment (RC2) · Anonymous Referee #2 · 25 May 2017

Summary:

This manuscript reports results of rBC emissions and emission ratios from two different agricultural fuel types, which can be emitted during open agricultural burning and are relevant to both China and other regions where wheat and rapeseed are grown worldwide. The authors do a good job of citing previous work on ambient rBC emissions and the large uncertainties and variation in the data from different locations and sources. Unfortunately this work only focuses on rBC and does not measure non-rBC mass or aerosol optical properties. A large source of uncertainty in the aerosol optical properties from OBB is due to the presence of BrC as well as rBC, which the authors acknowledge in the introduction, but do not attempt to measure or quantify. For example, even with the assumption that the non-rBC aerosol is dominated by organics (and not direct measurement), this work could predict total aerosol optical properties and BrC absorption using the Saleh et al. 2015 (already cited) and Pokhrel et al. 2017 correlations of mass ratios with measured aerosol optical properties from different fuel types during a similar laboratory study on different biomass burning fuel emissions. A measurement of the total aerosol, Organic Aerosol and/or non-refractory aerosol mass to report rBC/OA or rBC/Total aerosol mass is used to bound BrC as referenced above as well as the total aerosol optical properties (single scatter albedo, SSA) that has been found to also be independent of fuel type and as a function of MCE. SSA is relevant to the total aerosol radiative impacts of OBB as reported Liu et al., 2014. For all of these reasons, the addition of non-rBC measurements if available for this data set would greatly enhance the impact of this work on the total aerosol optical and physical properties from near-field source emissions of two major crops from China, wheat and rapeseed, and any attempt to add this kind of total aerosol information if possible would be greatly supported.

References: Saleh et al. Contribution of brown carbon and lensing to the direct radiative effect of carbonaceous aerosols from biomass and biofuel burning emissions. JGR-Atmospheres, 2015.

Pokhrel et al. Relative importance of black carbon, brown carbon, and absorption enhancement from clear coatings in biomass burning emissions. ACP, 2017.

Liu, S. et al. Aerosol single scattering albedo dependence on biomass combustion efficiency: Laboratory and field studies. GRL, 2014.

General Comments:

The rBC sample was diluted by a factor of 46 while the gas-phase measurements were diluted by a factor of 22. There a concern that such a large dilution of the rBC would make quantification of the total rBC mass from the fire emissions have very large uncertainties in the measurements. A study of the uncertainties induced by the

dilution system was studied for the aerosol sampling, but was not quantified for the gas-phase measurements. Are the authors not concerned that the different dilution ratios for the aerosol measurements and the gas-phase measurements might not introduce uncertainties in the measurements as the emissions ratios are the main deliverable of this manuscript? It is also unclear as to why the authors did not conduct dilution studies to see if the rBC coating was changing by introducing a 1/46 dilution ratio. Sampling a range of initial fuel sample emissions including smaller burn sizes (< 20 g) from the same fuel types would have greatly enhanced this study.

What is the width of the rBC size distributions from each experiment? While MMD of rBC MED is reported for each experiment, for example, what is the sigma or range in rBC size distributions? Is the rBC a tight size distribution/at what diameter does the rBC drop off for both the high and low ends? A table with this information and/or a graph of the rBC size distribution averages or examples would make a good addition to this manuscript in understanding the size range of rBC emissions.

Could the data from each burn be separated into smoldering and flaming analysis? What was the reason for using a fire-integrated MCE analysis when the first Figure separates the different phases? How was the separation of combustion conditions done for that figure? Was it using 0.9 as referenced in the introduction (Page 3 Line 10-11) or 0.95 in the results section (Page 11 Line 2) to separate the phases (or something else)? This information is referenced in different ways in the results section, but should be clearly defined in the experimental section and remain constant throughout the results section (which it may be but it's not clear to the reader how this was done).

In the absence of other size distribution or measurements of the non-refractory or scattering aerosol, if this SP2 is able to measure scattering particles up to 1 micron in diameter, could the scattering data be presented in addition to the rBC data to give a more representative picture of the total aerosol emissions and optical properties?

Similar to the response of the first reviewer, the authors focus on the combustion state

influencing the rBC emissions. What about the effect or concern for differences due to different fuel types, e.g. agricultural fuels versus wildfire fuels? Fuel types vary largely for OBB, and this should not be neglected. The authors are advised to modify the interpretation of the results and text at times to accommodate this as another reason for the large variability in reported rBC emissions from both laboratory and ambient measurements.

The addition of the wet data needs further substantiation in the methods, focus in the text, and data interpretation. Without this it should be removed from the text (or alternatively moved to SI).

Specific Comments: Page 1, Line 14: Are "rape plants" the same as rapeseed? If so, suggest adding "also known as rapeseed" to the text.

Page 1, Line 15: Do the authors mean "used" when they say "adopted"? Adopted implies a change was made to the standard SP2 rBC sampling regime. If this was done, please state and explain, and if it was not, please change the text to "used" or similar wording as the SP2 is a standard instrument for measuring rBC.

Page 1, Line 1 – Page 2, Line 1: "This study highlights that open biomass burning produces the majority of coated rBC particles, which have considerable ability to affect cloud processes and influence regional climate." This significance statement in the abstract overstates the results reported in this paper. It is unclear how the authors can state that biomass burning produces the majority of rBC coated particles in the atmosphere from a laboratory study of two different agricultural fuel types. A similar statement could be made along the lines of agricultural fuel types produce coated rBC particles, . . .", which would not over interpret the reported results.

Page 2, Line 3-4: What about cloud albedo?

Page 2, Line 7: It would be helpful to define OBB since this is not common terminology for a general audience. The reviewer suggests defining OBB to include agricultural and

wildfire emissions, but mainly suggests adding a sentence to define OBB clearly to the reader.

Page 2, Line 9: Do the authors mean VOCs or SVOC's? Both are common terms and are not used interchangeably.

Page 2, Line 10: Suggest removing "in smoke"

Page 2, Line 19 – 20: "Hygroscopic growth of rBC-containing particles also results in much more compact rBC cores." Is there a reference to support this statement? Suggest moving the Fan 2016 reference to a modelling study in the following sentence here and at the beginning of the sentence adding "Modelling studies indicate that the . . ." unless a measurements reference can also be added to support this statement.

Page 2, Line 21-22: "Second, the rBC particles are often located at off-center positions or may possibly be attached to the surfaces of non-rBC particles." Are there any references that can substantiate this sentence? If none can be found, please remove this sentence or change it to a statement implying these morphologies are possible but not implying the significance of off-center rBC particles in ambient aerosols.

Page 2, Line 28: add Liu 2015b to the list of citations for BrC influencing overall rBC absorption enhancements.

Page 3, Line 4-5: Suggest relating tar balls to secondary organic aerosol SOA from BB sources to link the two terminologies. Are tar balls one type of SOA defined as being low volatility and from BB sources? Are there any known optical properties that can be ascribed to this particle type, e.g. likely to contain BrC? A brief summary/explanation of the definition of what a tar ball is in terms of its formation, sources, physical and optical properties would benefit a larger audience.

Page 3, Line 12: When defining the rBC emission ratio, rBC is rBC mass concentration, correct? Likewise CO is a mixing ratio? Suggest adding this information to the initial definition here.

Page 3, Line 27: "The variability in $\Delta rBC/\Delta CO$ among observational studies also result from differences in sampling locations and conditions." After this section would be a good time to introduce the topic of variability due to fuel type as suggested in the general comments.

Page 4, Line 19 – 20: Suggest removing "to preserve its original state" since while this storage would limit deposition onto the samples it would not preclude semivolatile evaporation and/or water loss etc.

Page 4, Line 26: "To monitor the evolution of the combustion process of biomass, the mixing ratios of $CO_2$ and $CO$ in the OBB smoke were measured simultaneously." The gas phase mixing ratios were measured "to monitor the combustion conditions" of the rBC, correct? The statement here seems to imply aging, which these experiments are more representative of near-field emissions and do not probe aerosol aging in the plume as might be interpreted with the current sentence. The reviewer also cautions the authors to imply that all fires proceed from flaming to smoldering combustion conditions both here and other locations in the text since OBB can vary over time and does not always proceed as straightforward as a laboratory study.

Page 5, Line 31-33: Suggest showing at least an example of the rBC size distribution from one or an average from several burns ideally as a figure in the main text, and alternatively in the SI. Is a Gaussian fit best or does lognormal fit the rBC mass equivalent diameter data better?

Page 6, Line 3-4: The SP2 scattering channel was not saturated for the 500 nm and 1000 nm PSL's? The lower limit of the scattering detection is listed as 166 nm. What is the upper limit for this instrument? If this SP2 can detect scattering particles up to 1 micron in diameter, it would advantageous to report this data in addition to the rBC measurements.

Page 8, Line 10: Fire-integrated MCE's are reported and listed in Table 2. What is the variability over the course of each burn? Could the range of MCE's from each

experiment also be included in this table?

Page 9, Line 8-10: Could this be related to how the burns were started? Information on the fires were started/lit should be added to the information in the experimental section as well as being considered as an potential explanation for this initial rBC peak in number at the start of sampling.

Page 9, Line 22: "... because the combustion process differed significantly." Does this imply that the experiments do not generally proceed from flaming to smoldering as well as the examples in Figure 1? Please explain what this sentence means as it was not clear to the reviewer.

Page 9, Line 24: "45 times dilution". Earlier this was stated as 46 – please explain the reason for the difference.

Page 9, Line 27-29: Reference is made to the rBC displaying "a perfect Gaussian distribution for all burning cases." Reference is also made to a Figure 2a, which appears to not exist in this version of the text. This size distribution information should be added to the Figure. Is the rBC distribution averaged over the course of the whole experiment? rBC distributions are not often perfect Gaussians, therefore, the addition of this information to the Figure should be included.

Page 9, Line 31: Change " tends to produce larger particles" to "tends to produce larger rBC particles".

Page 10, Line 2: "small" seems to contract the data in Figure 2 and previous sentence since the rBC MMD increases for flaming combustion.

Page 11, Line 1 – 4: But are these reported differences in delta rBC/delta CO statistically significant? Based on the uncertainties, there does not look to be enough difference within the uncertainties of the measurements to warrant significant difference and subsequent interpretation.

Page 11, Line 29-30: What is the reason for focusing on the time delay analysis data

when the LEO-fitting coating thickness analysis that yields more information with fewer uncertainties was also extensively done? Was the LEO-fitting analysis only done on the MED = 200 rBC core-containing particles? 200 nm MED is relatively large for most rBC studies, and even for some of the data presented here where MMD's are reported down to 144 nm MED rBC for some of the experiments. How much do the results change if 150, 180 or 220 MED cores were used for the LEO analysis?

Page 12 and Figure 4: Can you separate the data into flaming and smoldering to substantiate the claim that the two modes present in all the data are due to the different combustion phases?

Page 13, Line 6-13: Add the range of S/C reported for the ARCTAS data to the text. How much weight can be placed on a S/C change of 1.2 to 1.4? Could some of the other studies help to substantiate why this is a significant difference? More explanation and reference to other datasets here in the text would be advised since the data in Figure 5 appears to have a lot of scatter in the data and poor r2 fit values.

Page 13, Line 14 – 15: Is this for all the data? Is there a difference in the different fuel types or MCE flaming versus smoldering conditions if the data were to be averaged from all experiments and separated into different categories? Adding the range of thicknesses sampled should also be added to the text here as the coating thicknesses look to cover the full ranges reported by the aircraft data referenced in the text.

Page 13, Line 19 – 29: Since Figure S4 indicated coating thicknesses of 0 – at least 60 nm sampled from this data, is it possible to state that atmospheric aging results in increased rBC coatings? The data presented here and in Table 3 is from a large variety of fuel types, combustion conditions, and atmospheric aging timescales that this level of interpretation requires more investigation isolating different fuel types and atmospheric aging timescales.

Page 14, Line 4-5: The S/C ratio appears to increase with the EF of the NMVOC's for dry data while the wet wheat S/C does not look to depend on EF of NMVOC's. It also

looks as if the S/C for the wet data is at the maximum for the dry data. More discussion of these differences should be included in the text if this is found to be significant. If not, then the wet data should be removed from the manuscript as it does not have much interpretation of the data collected here anywhere else in the manuscript.

Page 14, Line 9-10: Without a reference for this hypothesis or substantiation from the data presented here, this should be removed as it is too speculative.

Technical Corrections: Page 2, Line 3: remove "the" from ". . . play a vital role in the climate change. . ."

Page 2, Line 5: remove "their" from "its their"

Page 2, Line 11: remove ", evidently"

Page 2, Line 19: remove "much"

Page 3, Line 27: "The variability in $\Delta$rBC/$\Delta$CO among observational studies also result from differences in sampling locations and conditions." – change "result" to "results"

Page 4, Line 10: Change ". . . we conducted open burning experiments. . . " to ". . . we conducted laboratory burning experiments. . . ".

Page 4, Line 20: Remove "generally"

Page 5, Line 32: change "fitted" to "fit"

Page 13, Line 9: Remove "As a matter of fact"

Page 14, Line 5: Change "open" to "laboratory"

Page 14, Line 13: Remove "urgently"

Page 14, Line 26: Remove "obviously"

Page 14, Line 27: Add "rBC" to say "result indicated that the rBC coagulation/growth. . ."

Table 1: Change "C/S ratio" to "S/C ratio"

[Figure]

Table 2: Should the last column say "MED of rBC MMD"?

Figure 1: Needs the information added to the figure or figure text on the different color blocks of data shown in yellow, red and blue.

Figure 2: Needs an explanation of the lines presented in the figure and the circle around one set of data. Also the text refers to 2a and 2b within the Figure which are not present.

Figure 4: Would changing the color scale on the number of rBC in the Figure enhance the ability for the reader to discern the two modes that can be separated with the histograms? Label the modes flaming and smoldering on the figure would also make the main points of the Figure more clear to the reader.

---

## Author Response (AR1)

**Reply to the comments of anonymous reviewer #1 on manuscript entitled " Emission characteristic of refractory black carbon aerosols in the fresh Asian biomass burning: a perspective from laboratory experiment "**

We appreciate very much the insight comments and recommendations of the reviewer in improving this paper and our future research. Here, we will response to all the comments one by one as follows:

*General comments:*

*The present manuscript describes a series of laboratory experiments on wheat straw and rape or rapeseed plants to quantify emission ratios and mixing state of refractory black carbon (rBC) using the Single Particle Soot Photometer (SP2). The ultimate objective of this study is to provide data that can augment field measurements of biomass burning (BB) events as well as providing important BB inventory used by models. The primarily findings of value to the community include the quantification of the delta-rBC/delta-CO ratio as a function of MCE (modified combustion efficiency); dependence of rBC mass mode diameter on MCE; and dependence of rBC mixing state on MCE for the two fuel sources studied. The biggest disappoint of this study is the lack of measurements of the rBC optical properties as a function of MCE. From a climate forcing point-of-view, quantification of BB optical properties is central to bounding the contribution of these events to aerosol radiative forcing. Consequently, this study is very myopic - only two fuels are examined and the experiments carried out are rBC- centric. This being said, the data is expected to find value in emission inventories and thus should be considered for publication after comments, listed below, are addressed.*

Reply: We fully agree with the reviewer's suggestions, that measurement of optical properties of rBC as a function of MCE will benefit the better understanding aerosol radiative forcing effect of biomass burning, because rBC particles that produced at different combustion phase of biomass mass have distinct light-absorbing capacity due to coating of non-refractory matters. At present the observational studies for the absorption enhancement effect of rBC-containing particles are still scant and conflicting. To clarify its radiative effect, more observational and experimental results are needed. Open biomass burning (OBB) is one of great important sources of rBC. In particular in East China, substantial amount of crop residue were unorganizedly burned in the field, which lead to degradation of air quality and serious health problems. Better constraining the emission factor of rBC of OBB plays a key role in reducing the uncertainty of emission inventory and improving simulations of Chemical Transport Model (CTM).

The initial idea of this study is to investigate the emission ratio of rBC ($\Delta$rBC/$\Delta$CO) from open burning of residues of two major economic crops (wheat and rape plant) in East China (including Jiangsu, Anhui and Shandong province), because intensive burning of these crop residues in the field often resulted in Air Quality Index > 500 (hazardous level) during harvest season, which imposed great detrimental effect on human and environment. Long-range transport of Asian biomass burning was also frequently reported in Japan. For this reason, a Japan-China joint field campaign was performed at agriculture area in the East of China, we collected samples of crop residues and performed burning experiments in the laboratory. Taking advantage of SP2, the incandescent delaytime and coating thickness were also investigated to understand the physical characteristics of freshly emitted rBC particles. As reviewer pointed out, simultaneous measurements of optical properties and mixing state of rBC particles as a function of MCE was critically important to evaluate its climate effect. Liu et al., (2014) reported aerosol single

scattering albedo (ω) dependence on biomass combustion efficiency in laboratory and field study. The author reported that MCE could explains 60% of the variability in ω, while the 40% unexplained variability could be accounted for by other parameters such as fuel type. Besides, for open burning of agriculture residues in the field, the plumes were quickly mixed and diluted after being emitted in the ambient condition. rBC particles tend to be coated with more non-refectory matter (semi-volatile organics, inorganics etc.) with photochemical aging. Consequently, chemical and optical properties of rBC particles from burning may change during transport, whereas $\Delta rBC/\Delta CO$ ratio could preserve for long. As far as I know, a photo-acoustic soot spectrometer (such as PASS-3, DMT) could measure scattering/absorption coefficient at three wavelengths, and it was normally used to investigate the light absorption enhancement by mixed condition black carbon particles (Liu et al., 2014, Liu et al., 2017). Unfortunately, we did not have it during experiment. For all these reason, concurrent measurement of optical properties of rBC-containing particles in this study was not conducted in this study. In our next research, we would like to follow reviewer's suggestions to perform comprehensive measurements on the dependence of both mixing state and optical properties of rBC particles on the combustion state of biomass burning.

Liu, S., et al. (2014), Aerosol single scattering albedo dependence on biomass com- bustion efficiency: Laboratory and field studies, Geophys. Res. Lett., 41, 742–748, doi:10.1002/2013GL058392.

Liu, D., et al. (2017) Black-carbon absorption enhancement in the atmosphere determined by particle mixing state, Nature Geosci, advance online publication, 10.1038/ngeo2901.

*Specific comments:*

*The authors correctly indicate that the combustion process is the driving force that dictates variations in emission characteristics. It is clear that the authors characterized stages of the burn as either flaming or smoldering, yet offer no boundary conditions as to when a burn was flaming vs smoldering. If MCE was used, what value determined if the data points were from an active flaming condition or smoldering condition? And, if as the authors point out, both stages could occur at the same time. On page 9, line 13, the authors state that "when the combustion shifted from the flaming dominant to the smoldering-dominant state. . ." what is the criteria used to characterize one stage over the other?*

Reply: We will clarify this point in the revised manuscript. In this study, the MCE value = 0. 95 was deemed as a criteria to distinguish flaming- dominant and smoldering-dominant combustion basically. Nevertheless, a conservative criteria (MCE < 0.90) was used to indicate smoldering-dominant combustion, in accordance with our previous study (Kondo et al., 2011). During the experiments, we found a prominent phenomenon that occurrence of peak of number concentration of non-rBC particles was obviously later than that of rBC particles (as shown in Figure 1), and the MCE value of ~0.95 normally fall in the middle of these two peak. It can be an indicator that the dominant combustion phase shifted from the flaming to smoldering. As mentioned, the mass of each sample was ~20 g, the combustion period was normally short that both flaming and smoldering stages sometimes could occur at the same time at the different part of biomass. To reduce such uncertainty, we used a fire-integrated increment of mixing ratios of CO and CO2 and an averaged MCE of each combustion cases to represent the dominant combustion phase in the following discussion. We will clearly state the MCE criteria in the revised manuscript.

Kondo, Y., et al. (2011), Emissions of black carbon, organic, and inorganic aerosols from biomass burning in North America and Asia in 2008, J. Geophys. Res., 116, D08204, doi:10.1029/2010JD015152.

*This reviewer is rather surprised that the authors fit the rBC size/mass distribution data to a Gaussian (page 1, line 19; page 9, line 27) as opposed to a lognormal. It is well-known that most aerosol distributions are skewed (e.g., exhibit a long tail at larger sizes) and thus are better described with a lognormal function (Hinds 1999). On page 9, line 28, the authors make reference to figure 2a that presumably shows an example dM/dlogDp plot. Such a plot does not exist. The authors are strongly encouraged to add this figure along with a lognormal fit.*

Reply: We consented to the reviewer's comments, and checked both Lognormal fit and Gaussian fit of mass size distribution of rBC particles (Normalized dM/dlogDp v.s. Mass Equivalent Diameter of rBC) for all the combustion cases. As suggested, the aerosol distributions are normally skewed. Indeed, lognormal function fitting is slightly better than that of Gaussian function in most instances. This phenomena is more prominent in smoldering-dominant case that that in flaming-dominant cases (as shown in Figure 1). We will correct the expressions in the revised manuscript, and add dM/dlogDp plot in the Figure 2a.

[Figure]

Figure 1 Gaussian and lognormal fitting for normalized mass size distribution of rBC for flaming- (left) and smoldering- dominant case (right).

*Central in their study is the use of MCE. The authors are encouraged to read the 2016 publication by Collier et al., (Regional Influence of Aerosol Emissions from Wildfires Driven by Combustion Efficiency: Insights from the BBOP Campaign; (2016) Environ. Sci. Technol.50, 8613−8622) with specific attention to Figure 4). While the Collier paper focuses on wildfires, the dependence of aerosol emissions on MCE the authors might this study relevant to theirs.*

Reply: As mentioned by the reviewer, our main conclusion is consistent with Dr. Collier's results that characteristics of biomass burning aerosols are strongly dependent on the MCE. As a matter of fact, in this study we found that the ratio of number concentration of non-rBC (w/o incandescent signal, such as semi-volatile organic matter) to that of rBC particles also showed an obviously increasing tendency with decrease of MCE, implying of high emissions potential of OA in smoldering combustion. It was also consistent with Dr. Collier's study. We would like to cite Dr. Collier's research to support our conclusion in the revised manuscript.

*Page 1, Line 15: The authors cite in their abstract that "A single particle soot photometer (SP2) was adopted to measure rBC-containing particles at high temporal resolution and with high*

*accuracy," yet do not explicitly discuss what was "adopted" to realize the high temporal resolution and high accuracy. If the authors altered some hardware/software aspect of the SP2 that improved upon its "out-of-the-box" capabilities, then they should explicitly discuss those changes. If nothing was not done, than eliminate this statement as it is misleading.*

Reply: Since we did not alter hardware/software aspect of SP2, we would like follow the reviewer's advise and remove the statement "at high temporal resolution and with high accuracy ".

*Page 4, line 19: The authors write " All of the biomass was stored in sealed plastic bags to preserve its original state." Sealing a sample will not prevent loss of semivolatile materials. Have the authors accounted for this or attempted to quantify this?*

Reply: We agree with the reviewer's comments that the semi-volatile matters in the fresh biomass will loss even they were sealed in hermetic bag. In this study, the agriculture residues that were collected in the field were almost dry (Figure 2). In northern and eastern China peasant usually dry their crop residues first and burn them intensively. For the dried biomass, the loss of semi-volatile matter should be negligible, compared with vigorous combustion processes. In this study, we did not measure the water content and carbon content. However, we would like to follow reviewer's suggestion to consider the influences of physical properties of biomass on the emission characteristics of rBC in our next study.

[Figure]

Figure 2 Photo of dry rapeseed plants that were collected in the field.

*Page 4, line 22: "flexible rubber hose". This reviewer assumes that the authors mean "conductive" tubing. If so, please state that.*

Reply: In this study, we use conductive silicon tube for sampling flow and measurement. To avoid overloading of aerosol particles in the heat-resistant combustion box (~144 L), the smoke was ventilate from top of the combustion box to outside of laboratory through a flexible rubber tube (OD: ~ 15cm) at a high flow rate (120 m3/h). Figure 3 is the diagram of the biomass burning experiment, described in previous literature: Inomata, S., Tanimoto, H., Pan, X., Taketani, F., Komazaki, Y., Miyakawa, T., Kanaya, Y., and Wang, Z.: Laboratory measurements of emission factors of nonmethane volatile organic compounds from burning of Chinese crop residues, Journal of Geophysical Research: Atmospheres, 120, 5237-5252, 2015.

[Figure]

Figure 3. A schematic diagram of the laboratory biomass burning experiment

*Page 4, line 24/25: The authors indicate that four samples were placed in humid conditions for 30-minutes to absorb moisture. How was the moisture content quantified? Why only 30-minutes? What was the goal? To 'coat' the fuel with some moisture or increase the moisture content of the fuel? The moisture content would be expected to potentially impact the MCE and, in turn, the rBC/CO ratio and thus better quantification would be warranted.*

Reply: In this study, we prepared only four wet samples for investigating the impact of physical condition (dry or humid) of biomass on the emission characteristics of rBC particles. Because all the biomass samples were dry, we did artificial treatment on the samples by exposing them in humid condition. The biomass moisture reached about 96% as determined from one measured sample (Inomata et al., 2015). During the burning experiments, relatively more smoldering condition were achieved for the wetted samples with averaged MCE values of 0.88, 0.90, 0.91 and 0.96. We found that emission of both rBC and NMVOCs were suppressed. In our next study, we would like to measure the moisture content of biomass to quantitatively investigate the effects.

*Page 7, line 14/15: The authors assume that the non-refractory coating possesses a refractive index of 1.5 - 0i. While likely valid, the authors are encouraged to acknowledge that while BB events are a major source of brown carbon (BrC), it is highly unlikely that shortwave light absorbing OA will absorb at 1064 nm - the laser wavelength utilized by the SP2.*

Reply: Great thanks for the reviewer's reminds and suggestions; we will keep it in mind.

*Page 9, line 28/29: As stated above, Figure 2a does not exist.*

Reply: We will add the figures in the revised manuscript.

*Page 10, line 25 - 28. While some trends appear to be present, the lack of water content quantification limits how much can be concluded with respect to comparing dry and wet wheat straw. The authors are encouraged to address this either by estimating the change in water*

*content that a 30-minute exposure of a 99% RH environment could create or acknowledge that the lack of water content quantification limits the quantitative comparison of dry and wet wheat straw emission ratios.*

Reply: We would like to follow the reviewer's advise to make it clear in the manuscript that the burning experiment on wet biomass only provide quantitative results because of lack of water content quantification of biomass, although $\Delta rBC/\Delta CO$ ratio for burning of wet biomass was observed to be lower than that of dry biomass. Previous study (Chen, L. W., et al., Moisture effects on carbon and nitrogen emission from burning of wildland biomass, Atmospheric Chemistry and Physics, 10, 6617-6625, 2010.) using off-line filter-based analysis (IMPROVE_A protocol) indicated that emission factor (g/KgC) of EC did not change or even decrease as increase in moisture level of biomass. Their result was consistent with ours. We would like to quantify the relationship between water content and emission characteristics of rBC in our next research.

*Page 11/12 and Figure 4. The linearity between the incandescence delay time and shell/core ratio is somewhat surprising. In the application and comparison of the two methods of analyzing the rBC mixing state - incandescence delay and coating thickness was the LEO method applied to the incandescence delay time analysis? Not only will the LEO method impact the scattering signal amplitude, it could impact the scattering peak location relative to the rBC incandescence peak. Therefore, as the shell/core ratio increases evaporative losses might be expected to exert a greater impact the location of the uncorrected and LEO corrected delay times.*

Reply: Yes, as pointed out by the reviewer, the S/C ratio in Figure 4 was calculated on the basis of LEO method only for the rBC-containing particle with delay time ($\Delta t$) > 0.8 μs, and the rBC particles with negative delaytime was excluded in LEO fitting analysis because in such cases rBC particles may locate at off-center positions or attach to the surfaces of non-rBC matters. To be clear that, LEO fitting method was adopted to estimate the original scattering signal amplitude of rBC-containing particle before the evaporative loss started. Whereas, $\Delta t$ is observed difference in scattering peak location and incandescence peak location. They are independent parameters. To derived the rBC-containing particles' original scattering signal amplitude, LEO fitting method just needs two basic information (1) the position of center of the laser beam in the observed scattering signal profile, and we determined this position on the basis of SP2's Position Sensitive detector (PSD, Gao et al., 2007) and standard procedures (Laborde et al., 2012); Briefly, it was calculated using equation: $t_{split} + \Delta t_{split-center}$. Here, $t_{split}$ is the observed position that PSD scattering signal (Channel 3) was inverted, and $\Delta t_{split-center}$ is a predetermined parameter describing time difference between center of laser beam ($t_{center}$) and $t_{split}$ that was obtained on the basis of PSL experiment. (2) "leading edge" data that were used for Gaussian fitting. In this study, the leading edge data are selected according to the criterion of t < -2.5σ. Here, σ denotes the standard deviation of the Gaussian function of the laser intensity profile, as described in literature (Moteki et al., 2014). Therefore, the LEO method has nothing to do with scattering peak location relative to the rBC incandescence peak.

Regarding the linearity between S/C ratio and $\Delta t$, we are sorry for the misleading of the statement in the manuscript. Figure 4 mainly describes the relationship between S/C ratio and $\Delta t$ for rBC-containing particles with rBC cores (MED = 200 ± 10 nm). As shown, the histograms of both the S/C ratio and $\Delta t$ showed a predominant peak with a relatively long tail. A multiple-peak Gaussian fitting analysis showed that there were two modes with S/C ratio = 1.18 and S/C ratio = 1.34, corresponding to $\Delta t$ = 1.74 μs and $\Delta t$ = 3.18 μs. It reflected that the rBC particles had different levels of coatings in different combustion state. Meanwhile, $\Delta t$ value for biomass burning aerosol was larger than that of ambient EC in the suburbs of Tokyo (Moteki et al., 2007)

at the same S/C ratio. This phenomenon was possibly related to both chemo-physical properties of coating matters and irregularity of rBC particles. Further experiments are need.

*Page 13, line 19/20: The authors state that "The coating thickness of freshly emitted rBC particles from OBB was relatively small (20 nm), and this thickness was reported to increase to 65 ± 12 nm (Schwarz et al, 2008) and up to 100 nm (Taylor et al., 2014) when they experienced transport over hours or days." The authors are cautioned here. As Schwarz et al. state: "Although the sources of these emissions are unknown, their location and season of occurrence suggest that neither BB plume is from agricultural sources, but from brush fires." Similarly, Taylor et al., interrogated a boreal forest fires. The source fuel examined by the authors is agricultural in origin, not wilfires. Therefore caution must be exercised when extrapolating to expected aging behavior using two very different source fuels. As a matter of fact, this Reviewer is not convinced of the statement "We found that the aging of particles was more important than their sources in determining the coating thickness of rBC particles." More discussion is needed to buttress this statement.*

Reply: We consent to the reviewer's comment that it was hardly to attribute the difference in coating thickness of rBC particle among studies only to atmospheric aging because fuel types, combustion condition were different. We will add type of biomass in the Table 3 and revise the interpretation in the revise manuscript. Besides, we noticed that, for the rBC particles outflowed from urban area, the number fraction of thickly-coated rBC particles (S/C ratio > 2 @ rBC$_{MED}$ = 180 nm) could increased from 30% to 60% of total rBC particles as their photochemical age increased from 2h to 14 h (Moteki et al., 2007). It reflected that the photochemical process was of great importance in variation of the mixing state of rBC particles. At present, more field/airborne observational evidences of biomass burning aerosols were needed to explain the variability in coating thickness of rBC particles.

*Please insert error bars on Figure 6 if possible*

Reply: We will add error bar in the Figure 6.

[Figure]

Figure 6. Variations in the shell/core ratios of rBC particles with MED = 200 ±10 nm as a function of the emission factor of each experiment. Here, EF is defined as the amount of each compound released per unit amount of dry fuel consumed. The red, green and blue colors indicate the dry wheat straw, wet wheat straw and rapeseed plant samples, respectively.

**Reply to the comments of anonymous reviewer #2 on manuscript entitled " Emission characteristic of refractory black carbon aerosols in the fresh Asian biomass burning: a perspective from laboratory experiment "**

We appreciate very much the insight comments and recommendations of the reviewer in improving this paper and our future research. Here, we will response to all the comments one by one as follows:

*This manuscript reports results of rBC emissions and emission ratios from two different agricultural fuel types, which can be emitted during open agricultural burning and are relevant to both China and other regions where wheat and rapeseed are grown worldwide. The authors do a good job of citing previous work on ambient rBC emissions and the large uncertainties and variation in the data from different locations and sources. Unfortunately this work only focuses on rBC and does not measure non-rBC mass or aerosol optical properties. A large source of uncertainty in the aerosol optical properties from OBB is due to the presence of BrC as well as rBC, which the authors acknowledge in the introduction, but do not attempt to measure or quantify. For example, even with the assumption that the non-rBC aerosol is dominated by organics (and not direct measurement), this work could predict total aerosol optical properties and BrC absorption using the Saleh et al. 2015 (already cited) and Pokhrel et al. 2017 correlations of mass ratios with measured aerosol optical properties from different fuel types during a similar laboratory study on different biomass burning fuel emissions. A measurement of the total aerosol, Organic Aerosol and/or non-refractory aerosol mass to report rBC/OA or rBC/Total aerosol mass is used to bound BrC as referenced above as well as the total aerosol optical properties (single scatter albedo, SSA) that has been found to also be independent of fuel type and as a function of MCE. SSA is relevant to the total aerosol radiative impacts of OBB as reported Liu et al., 2014. For all of these reasons, the addition of non-rBC measurements if available for this data set would greatly enhance the impact of this work on the total aerosol optical and physical properties from near-field source emissions of two major crops from China, wheat and rapeseed, and any attempt to add this kind of total aerosol information if possible would be greatly supported.*

Reply: We consented to the reviewer's comments that open biomass burning (OBB) emits not only refractory BC (rBC) particles but also substantial mounts of organic aerosols (OA), the latter of which mostly consists of light-absorbing carbon (BrC). The chemical/physical and mixing state of BrC with rBC were not well understood, which resulted in great uncertainty in evaluating the climate effect of biomass burning aerosol. Recent studies (Saleh et al., 2014, Pokhrel et al., 2017) make advances in better quantify the absorptivity and direct radiative effect of BrC mass by introducing an applicable parameter (BC-to-OA ratio)(Saleh et al., 2015). They also revealed that absorptivity of BrC depend largely on its combustion condition at the source (Saleh et al., 2014, Collier et al., 2016), reflecting of the importance to investigate the optical properties of fresh biomass burning aerosols. In the present study, initial idea is to investigate the emission characteristics of rBC from open burning of agriculture residues in East China to better constrain uncertain of rBC emission inventory and to improve the performance of regional chemistry model, because intensive crop residues burning in the field often resulted in local Air Quality Index > 500 (hazardous level) in harvest season in East China. Because of the limit in instruments, the optical property of OBB aerosol was not simultaneously measured during experiments. Nevertheless, we would like to follow the reviewer's advice to incorporate optical measurements of OBB-related aerosols in our further research.

Regarding reviewer's suggestion to predict BrC absorption on the basis of assumption that the non-rBC aerosol is dominated by organics and BC-to-OA ratio. Note that, SP2 only counts the number concentration of light scattering particle or non-rBC particles with diameter larger than 166 nm due to limitation of avalanche photodiode detector, and the size of non-rBC particles are derived on the basis of predetermined calibration curve using polystyrene sphere latex particles (PSL, Size Standard Particles, JSR Corporation, Japan). Therefore, precise estimation on optical properties of organics aerosols was difficulty due to lacking the size-resolved density, morphology information. In particular, the information about existing state of non-rBC mass (internally mixed or external mixed with rBC) for the particle with a diameter less than 166 nm was also unknown. As a whole, it is better to predict the optical property of OBB aerosol on the basis of direct measurement. Now we are purchasing a centrifugal particle mass analyzer (CPMA), we will use a DMA-CPMA-SP2 tandem system to quantify the mass concentration of rBC and non-rBC, as well as the optical properties of rBC-containing particles. By selecting a specific particle size range (for example, 200 ~ 300 nm), the mixing state and absorptivity of BrC will be quantified properly.

*General comments:*

*The rBC sample was diluted by a factor of 46 while the gas-phase measurements were diluted by a factor of 22. There a concern that such a large dilution of the rBC would make quantification of the total rBC mass from the fire emissions have very large uncertainties in the measurements. A study of the uncertainties induced by the dilution system was studied for the aerosol sampling, but was not quantified for the gas-phase measurements. Are the authors not concerned that the different dilution ratios for the aerosol measurements and the gas-phase measurements might not introduce uncertainties in the measurements as the emissions ratios are the main deliverable of this manuscript?*

Reply: We agree with reviewer that different dilution factor could result in different uncertainties in both aerosol-phase and gas-phase measurement, and finally influences the emission ratio of rBC. During biomass burning experiment, we used a high dilution factor (~46) for aerosol channel to satisfy the maximum detection limits of SP2. As a matter of fact, the same dilution factor (46) was also examined for gas-phase measurement. We found that uncertainty (2%) of carbon monoxide measurement was negligible. Because an ultrafast CO analyzer (model AL5002, Aero- 30 Laser GmbH) has relatively high detection limit (detection range 0–100 ppmv, detection limit 1.5 ppbv, integration time 1 s), we selected a low dilution factor (22). Besides, the inlet of gas-phase and aerosol-phase sampling tube was ~40 cm beside the flame of biomass, therefore except for 46 times excess dilution, biomass smoke has already underwent great dilution at the sampling location due to mixing with excess fresh air. In this study, we considered that the emission ratio of rBC ($\Delta$rBC/$\Delta$CO) preserved because they were emitted in seconds and they did not experience in-cloud and below-cloud scavenging processes. In future work, we will follow reviewer's suggestion to use the same dilution ratio when both rBC and CO concentrations were measured.

*It is also unclear as to why the authors did not conduct dilution studies to see if the rBC coating was changing by introducing a 1/46 dilution ratio. Sampling a range of initial fuel sample emissions including smaller burn sizes (< 20 g) from the same fuel types would have greatly enhanced this study.*

Reply: We agree to the reviewer's suggestions. As implied, different dilution condition will impact the gas-aerosol equilibrium of semi-volatile organic matter, the latter of which tend to influence the mixing state of rBC particles due to continuous condensation/evporation

processes. In this study, we did not quantify the effect of dilution on the coating thickness of rBC particles because the total dilution factor can not be exactly determined since the sampling inlet was besides the combustion flame, and biomass smoke has already diluted due to mixing with fresh air in the combustion chamber. Besides, to avoid saturation of SP2 measurement, the burning biomass was only ~20 gram, and the rapid evolution of the combustion process also resulted in difficulty in discriminating the dilution effect. Therefore, large-scale burning experiment (like FLAME experiments) is much applicable for investigating dilution effects, and parallel measurements using two SP2 with different dilution factors is another option. In our future research, we would like to follow the reviewer's advice to investigate the dilution effect.

*What is the width of the rBC size distributions from each experiment? While MMD of rBC MED is reported for each experiment, for example, what is the sigma or range in rBC size distributions? Is the rBC a tight size distribution/at what diameter does the rBC drop off for both the high and low ends? A table with this information and/or a graph of the rBC size distribution averages or examples would make a good addition to this manuscript in understanding the size range of rBC emissions.*

Reply: Great thanks for the reviewer's suggestions. We will add the information in the table 2. Besides, We make a correction in the curve fitting of mass size distribution of rBC particles according to the comments of the reviewer #1. As a matter of fact, Lognormal function fitting is better than that of Gaussian function in most instances. This phenomena is more prominent in smoldering-dominant case that that in flaming-dominant cases (as shown in Figure 1). We will correct the expressions in the revised manuscript, and add dM/dlogDp plot in the Figure 2a.

[Figure]

Figure 1 Gaussian and lognormal fitting for normalized mass size distribution of rBC for flaming- (left) and smoldering- dominant case (right).

*Could the data from each burn be separated into smoldering and flaming analysis? What was the reason for using a fire-integrated MCE analysis when the first Figure separates the different phases? How was the separation of combustion conditions done for that figure? Was it using 0.9 as referenced in the introduction (Page 3 Line 10-11) or 0.95 in the results section (Page 11 Line 2) to separate the phases (or some- thing else)? This information is referenced in different ways in the results section, but should be clearly defined in the experimental section and remain constant throughout the results section (which it may be but it's not clear to the reader how this was done).*

Reply: As mentioned in the manuscript, the combustion processes evolved rapidly. Flaming and smoldering combustion sometimes occurred at the same time at different part of biomass, and clear discrimination between flaming phase and smoldering phase was difficult. Therefore, we used a fire-integrated MCE to represent the general burning condition of biomass. Regarding the criterion, a value of MCE > 0.95 and a value of MCE < 0.9 was used in literatures as criterion to separate flaming and smoldering phase, respectively. We followed the same criterion in the manuscript. Figure 1 shows two examples reflecting the temporal variations of number concentrations of rBC and non-rBC particles. We found a prominent phenomenon that occurrence of peak of number concentration of non-rBC particles was obviously later than that of rBC particles, and the MCE value = 0.95 always fall in the middle of these two peak. It can be an indicator that the combustion shifted from the flaming-dominant to smoldering-dominant state. We will clearly state the MCE criteria in the revised manuscript.

*In the absence of other size distribution or measurements of the non-refractory or scattering aerosol, if this SP2 is able to measure scattering particles up to 1 micron in diameter, could the scattering data be presented in addition to the rBC data to give a more representative picture of the total aerosol emissions and optical properties?*

Reply: As suggested by the reviewer, we would like to add size-distribution information about both the rBC and non-rBC particles during the burning experiment, as shown Figure 2 below. SP2 only detects light-scattering or non-rBC particles with diameter larger than ~ 166 nm. Presuming that non-rBC particles were in spherical configuration, the volume size distribution could be also determined. In the present study, non-rBC particles less than 166 nm in size (the lower detection limit of the SP2) accounted for limited fraction (4%) of the total aerosol volume. The maximum size of non-rBC particle was estimated to be ~ 750 nm. It is worth noting that we found that there were two modes for the volume size distribution of non-rBC particles with a peak at 307 nm and another peak at 606 nm. The second minor peak was related to an extreme smoldering case (discussed in paper, Inomata et al., 2015, JGR), since the number concentration was very low.

[Figure]

Figure 2 Normalized number size distribution (dN/dlogDp) and normalized volume size distribution (dV/dlogDp) for non-rBC particle during experiment.

*Similar to the response of the first reviewer, the authors focus on the combustion state influencing the rBC emissions. What about the effect or concern for differences due to different fuel types, e.g. agricultural fuels versus wildfire fuels? Fuel types vary largely for OBB, and this should not be neglected. The authors are advised to modify the interpretation of the results*

*and text at times to accommodate this as another reason for the large variability in reported rBC emissions from both laboratory and ambient measurements.*

Reply: We consented to the reviewer's insight comments. To avoid misleading, we will specify the type of biomass in the table 3 and correct our statements in the revised manuscript. As the reviewer implied, there are large differences in carbon/water contents and physical structure for different biomasses, which may result in their different inflammability, as well as mixing state of rBC particles, though some studies (e.g. Collier et al., EST, 2016) pointed out that general characteristics of biomass burning aerosols depended strongly on the combustion processes of a fire, and Saleh et al., (2014, NG) reported that aerosol absorptivity depends largely on burning condition, not fuel type.

*The addition of the wet data needs further substantiation in the methods, focus in the text, and data interpretation. Without this it should be removed from the text (or alternatively moved to SI).*

Reply: As suggested, we will move this part to supporting information.

*Specific Comments:*

*Page 1, Line 14: Are "rape plants" the same as rapeseed? If so, suggest adding "also known as rapeseed" to the text.*

Reply: We will use rapeseed plants in the context for consistency.

*Page 1, Line 15: Do the authors mean "used" when they say "adopted"? Adopted implies a change was made to the standard SP2 rBC sampling regime. If this was done, please state and explain, and if it was not, please change the text to "used" or similar wording as the SP2 is a standard instrument for measuring rBC.*

Reply: In the revised manuscript, we will replace "adopted" by "used".

*Page 1, Line 1 – Page 2, Line 1: "This study highlights that open biomass burning produces the majority of coated rBC particles, which have considerable ability to affect cloud processes and influence regional climate." This significance statement in the abstract overstates the results reported in this paper. It is unclear how the authors can state that biomass burning produces the majority of rBC coated particles in the atmosphere from a laboratory study of two different agricultural fuel types. A similar statement could be made along the lines of agricultural fuel types produce coated rBC particles, . . .", which would not over interpret the reported results.*

Reply: As suggested, we will revise the overstatement in manuscript, as follows: " This experimental study found that mixing state of rBC particles from biomass burning strongly depend on the its combustion processes, and overall MCE should be took carefully into consideration while the climate effect of rBC particles from open biomass burning was simulated."

*Page 2, Line 3-4: What about cloud albedo?*

Reply: We will add "cloud albedo" in the sentence.

*Page 2, Line 7: It would be helpful to define OBB since this is not common terminology for a general audience. The reviewer suggests defining OBB to include agricultural and wildfire emissions, but mainly suggests adding a sentence to define OBB clearly to the reader.*

Reply: we will add a terminology in the manuscript.

*Page 2, Line 9: Do the authors mean VOCs or SVOC's? Both are common terms and are not used interchangeably.*

Reply: Thanks for pointing out the typo, and I will delete the "VOCs" since it is not used in the context.

*Page 2, Line 10: Suggest removing "in smoke"*

Reply: "in smoke" is removed in the revised manuscript.

*Page 2, Line 19 – 20: "Hygroscopic growth of rBC-containing particles also results in much more compact rBC cores." Is there a reference to support this statement? Suggest moving the Fan 2016 reference to a modelling study in the following sentence here and at the beginning of the sentence adding "Modelling studies indicate that the . . ." unless a measurements reference can also be added to support this statement.*

Reply: We revised the sentence as suggested, as follows: " Modeling study indicated that hygroscopic growth of rBC-containing particles also results in more compact rBC cores (Fan et al., 2016). "

*Page 2, Line 21-22: "Second, the rBC particles are often located at off-center positions or may possibly be attached to the surfaces of non-rBC particles." Are there any references that can substantiate this sentence? If none can be found, please remove this sentence or change it to a statement implying these morphologies are possible but not implying the significance of off-center rBC particles in ambient aerosols.*

Reply: The sentence will be changed to "Second, the rBC particles sometimes were also reported to be attached on the surface of non-rBC matters (Moteki et al., 2014)."

*Page 2, Line 28: add Liu 2015b to the list of citations for BrC influencing overall rBC absorption enhancements.*

Reply: I will cite Liu et al., 2015b in the manuscript.

*Page 3, Line 4-5: Suggest relating tar balls to secondary organic aerosol SOA from BB sources to link the two terminologies. Are tar balls one type of SOA defined as being low volatility and from BB sources? Are there any known optical properties that can be ascribed to this particle type, e.g. likely to contain BrC? A brief summary/explanation of the definition of what a tar ball is in terms of its formation, sources, physical and optical properties would benefit a larger audience.*

Reply: In the revised manuscript, we will state that " It was reported that "tar-ball" that mostly consisted of BrC and secondary organic aerosols with low volatility were also emitted during smoldering combustion (Pósfai et al., 2004)."

*Page 3, Line 12: When defining the rBC emission ratio, rBC is rBC mass concentration, correct? Likewise CO is a mixing ratio? Suggest adding this information to the initial definition here.*

Reply: Initial definition will be added in the revised manuscript, as folows: "The rBC emission ratio ($\Delta$rBC/$\Delta$CO), which is defined as the enhancement of mass concentration rBC (in unit of ng/m3) with respect to its background versus that of CO (in unit of ppbv, parts per billion volume), ... "

*Page 3, Line 27: "The variability in $\Delta$rBC/$\Delta$CO among observational studies also result from differences in sampling locations and conditions." After this section would be a good time to introduce the topic of variability due to fuel type as suggested in the general comments.*

Reply:I will follow the reviewer's suggestion to revise the content of second paragraph in page3, as follows: "The rBC emission ratio ($\Delta$rBC/$\Delta$CO), which is defined as the enhancement of mass concentration rBC (in unit of ng/m$^3$) with respect to its background versus that of CO (in unit of ppbv, parts per billion volume), is an applicable indicator for constraining the rBC emission inventory for models (Pan et al., 2011). ==The variability in $\Delta$rBC/$\Delta$CO among observational studies mostly results from differences in measurement techniques, fuel type, and burning conditions==. For example, observations made onboard the NOAA WP-3D aircraft yielded $\Delta$rBC/$\Delta$CO values of 9 ± 2 ng/m$^3$/ppbv (Spackman et al., 2008) and 17.4 ng/m$^3$/ppbv (Schwarz et al., 2008) ==for brush fire plumes== during the TexAQS field campaign. Airborne observations on the NASA DC-8 aircraft indicated that the $\Delta$rBC/$\Delta$CO values were 8.5 ± 5.4 ng/m$^3$/ppbv for plume of ==boreal forest and agriculture fires in Asia==, and 2.3 ± 2.2 ng/m$^3$/ppbv for ==wildfire plume== in North America (Kondo et al., 2011). Observations using a multi-angle absorption photometer (MAAP, which employs the filter-based light absorption technique; here we consider BC instead of rBC) at mountain sites (30.16°N, 118.26°E, 1840 m a.s.l.) in South China yielded high $\Delta$BC/$\Delta$CO values (10 - 14 ng/m$^3$/ppbv) when the site was subjected to ==burning of crop residues== (Pan et al., 2011)."

*Page 4, Line 19 – 20: Suggest removing "to preserve its original state" since while this storage would limit deposition onto the samples it would not preclude semivolatile evaporation and/or water loss etc.*

Reply:As suggested, the statement "to preserve its original state" will be deleted. As a matter of fact, the farmer normally dries agriculture residues in the sun for days before the biomasses are burned. The samples that we collected in the field were dehydrated so that evaporation of semi-volatile and water vapor should be negligible.

[Figure]

Figure 3 Photo of dry rapeseed plants that were collected in the field.

*Page 4, Line 26: "To monitor the evolution of the combustion process of biomass, the mixing ratios of CO2 and CO in the OBB smoke were measured simultaneously." The gas phase mixing ratios were measured "to monitor the combustion conditions" of the rBC, correct? The statement here seems to imply aging, which these experiments are more representative of near-field emissions and do not probe aerosol aging in the plume as might be interpreted with the current sentence. The reviewer also cautions the authors to imply that all fires proceed from flaming to smoldering combustion conditions both here and other locations in the text since OBB can vary over time and does not always proceed as straightforward as a laboratory study.*

Reply: We agreed to the reviewer's comments. Here, mixing ratios of CO and CO2 were measured to calculate Modified Combustion Efficiency (MCE), the latter of which is a applicable metric to estimate the combustion phase of biomass. To avoid misleading, we will revise the statement in the revised manuscript, as follows: "As mentioned, MCE is a useful metric for describing the combustion phase of biomass burning, and the calculation of MCE requires simultaneous measurements of CO and CO2 concentration. Here, mixing ratio of CO2 was measured using a Li-7000 CO2 analyzer..."

*Page 5, Line 31-33: Suggest showing at least an example of the rBC size distribution from one or an average from several burns ideally as a figure in the main text, and alternatively in the SI. Is a Gaussian fit best or does lognormal fit the rBC mass equivalent diameter data better?*

Reply: We will add the mass size distribution of rBC particle in the manuscript. We found that lognormal curve fitting is slightly better than that of Gaussian function for rBC particles in most burning instances, in particular for smoldering-dominant case, as shown in Figure 4. We will correct the statements.

[Figure]

Figure 4 Normalized mass size distribution of rBC particles for all the burning cases.

*Page 6, Line 3-4: The SP2 scattering channel was not saturated for the 500 nm and 1000 nm PSL's? The lower limit of the scattering detection is listed as 166 nm. What is the upper limit for this instrument? If this SP2 can detect scattering particles up to 1 micron in diameter, it would advantageous to report this data in addition to the rBC measurements.*

Reply: Great thanks for the reviewer's suggestion. Regarding calibration, scattering signal for PSL particle at both Dp = 500 nm and Dp = 1000 nm were saturated for APD of SP2. Nevertheless, peak height of scattering signal for PSL particle at Dp = 500 nm could be estimated properly according Gaussian fitting by SP2. Unfortunately, it was not available for

PSL particle at Dp = 1000 nm due to strict data screening. Maximum size of non-rBC particle was estimated to be ~ 750 nm, and the We will report the information in the revised manuscript.

*Page 8, Line 10: Fire-integrated MCE's are reported and listed in Table 2. What is the variability over the course of each burn? Could the range of MCE's from each experiment also be included in this table?*

Reply: We will include 10th, and 90th percentile values for each experiment.

*Page 9, Line 8-10: Could this be related to how the burns were started? Information on the fires were started/lit should be added to the information in the experimental section as well as being considered as an potential explanation for this initial rBC peak in number at the start of sampling.*

Reply: We will add information about the ignition of biomass in the experimental section, as suggested.

*Page 9, Line 22: ". . . because the combustion process differed significantly." Does this imply that the experiments do not generally proceed from flaming to smoldering as well as the examples in Figure 1? Please explain what this sentence means as it was not clear to the reviewer.*

Reply: We will revise the expression to avoid misleading. In general, the combustion proceed from flaming to smoldering for most case, however the time duration of in flaming or smoldering were different. We will also add the total duration of combustion processes in the Table. Of course, we cannot avoid the situations that both flaming and smoldering combustion occurred at different position of fuel at the same time. Therefore, we considered using a fire-integrated MCE to present the overall combustion condition.

*Page 9, Line 24: "45 times dilution". Earlier this was stated as 46 – please explain the reason for the difference.*

Reply: We will correct the mistake.

*Page 9, Line 27-29: Reference is made to the rBC displaying "a perfect Gaussian distribution for all burning cases." Reference is also made to a Figure 2a, which appears to not exist in this version of the text. This size distribution information should be added to the Figure. Is the rBC distribution averaged over the course of the whole experiment? rBC distributions are not often perfect Gaussians, therefore, the addition of this information to the Figure should be included.*

Reply: As suggested, we will include the mass size distribution of rBC particles as shown previously.

*Page 9, Line 31: Change " tends to produce larger particles" to "tends to produce larger rBC particles".*

Reply: We will change to "tend to produce larger rBC particles"

*Page 10, Line 2: "small" seems to contract the data in Figure 2 and previous sentence since the rBC MMD increases for flaming combustion.*

Reply: Sorry for misleading, I will revise this statement as follows: " This result indicates that flaming combustion tends to produce larger rBC particles than smoldering combustion. It is consistent with previous studies, which reported that rBC particles formed considerably in intense flaming combustion due to less efficient transport of oxygen into the interior flame zone. As a result, growth in the size of rBC particles was rapid because the coagulation rate of particles is roughly proportional to the square of their number concentration (Lee and Chen, 1984)."

*Page 11, Line 1 – 4: But are these reported differences in delta rBC/delta CO statistically significant? Based on the uncertainties, there does not look to be enough difference within the uncertainties of the measurements to warrant significant difference and subsequent interpretation.*

Reply: We will add more information and revise the manuscript as follows: "the average $\Delta$rBC/$\Delta$CO ratio was $13.9 \pm 10.1$ ng/m$^3$/ppbv for the burning cases with a fire-integrated MCE > 0.95. This value was probably a low estimation since both flaming and smoldering combustions were included in the calculation. However, by selecting the cases with both the 10th and 90th percentiles MCE value > 0.90, we found that the average $\Delta$rBC/$\Delta$CO ratio was $23.1 \pm 11.4$ ng/m$^3$/ppbv ...".

*Page 11, Line 29-30: What is the reason for focusing on the time delay analysis data when the LEO-fitting coating thickness analysis that yields more information with fewer uncertainties was also extensively done? Was the LEO-fitting analysis only done on the MED = 200 rBC core-containing particles? 200 nm MED is relatively large for most rBC studies, and even for some of the data presented here where MMD's are reported down to 144 nm MED rBC for some of the experiments. How much do the results change if 150, 180 or 220 MED cores were used for the LEO analysis?*

Reply: As pointed out by the reviewer, LEO-fitting analysis could provide direct information about the coating thickness of rBC particle. Nevertheless, the delaytime of occurrence of incandescent peak *after* scattering peak also was informative for the reader to understand the real situation when the particle was passing through the laser. For instance, number of previous studies (Sedlacek et al., 2012, GRL; Moteki et al., 2007, GRL; Taketani et al., 2014, JGR, Miyakawa et al., 2015, AE, etc.) directly used the delaytime to indicate the degree of aging/coating of rBC-containing particles. rBC-containing particle in non-shell-core structure could also be determined if the delaytime value was a negative value (Sedlacek et al., 2012).

Regarding the LEO-fitting analysis, we agree with the reviewer's concerns that most of rBC particles had small size (less than 200 nm), and coating thickness could be calculated on the basis of LEO-fitting method. It was worthy noting that low detection limit of scattering particles was ~ 166 nm, implied that information about the rBC-containing particles with a diameter less than 166 nm was missing, and it will be problematic statistically to identify the relationship between coating thickness and MCE. Although we did analysis for all the data, we just reported the rBC particles with larger MED in the manuscript.

[Figure]

Figure 5 An example of variation of delaytime as a function of MED of rBC. Line-shaded area indicates that the information is missing. The yellow-color shaded area indicate the data that was analyzed in the manuscript.

*Page 12 and Figure 4: Can you separate the data into flaming and smoldering to substantiate the claim that the two modes present in all the data are due to the different combustion phases?*

Reply: In this study, we found that S/C ratio for rBC particles with MED = 200 nm increase as MCE decrease in most of burning case (Figure 6 shows two examples). As demonstrated, S/C ratio was ~ 1.2 when MCE > 0.95 (flaming-dominant phase) and ~ 1.4 when MCE < 0.90 (smoldering-dominant phase), in accordance with the estimated results from multi-Gaussian curve fitting. Because the data in flaming and smoldering phase overlapped significantly (as Figure 4 shown in the manuscript), it is better to demonstrate the differences using multi-Gaussian curve fitting method. We will add more information in the Figure 4 in the manuscript for better understanding of readers.

[Figure]

Figure 6 Two examples demonstrating that S/C ratio increase as MCE decrease for rBC particles with MED = 200 nm during combustion process.

*Page 13, Line 6-13: Add the range of S/C reported for the ARCTAS data to the text. How much weight can be placed on a S/C change of 1.2 to 1.4? Could some of the other studies help to substantiate why this is a significant difference? More explanation and reference to other datasets here in the text would be advised since the data in Figure 5 appears to have a lot of scatter in the data and poor r2 fit values.*

Reply: We will report the range of S/C ratio from ARCTAS data, and revise the statement as follows " Airborne measurements during the ARCTAS campaign (Kondo et al., 2011) showed an increasing tendency for the values of the S/C ratio (1.3 ~ 1.66) with increase of MCE, and the authors explained that this phenomena was because flaming phase plumes were more aged than the smoldering plumes, a 205 ± 40% increase of the volume of the coating materials resulted in a larger S/C ratio than that of rBC particles in smoldering plume."

We consented to the reviewer's comments that a S/C ratio increasing from 1.2 to 1.4 was not significant. As a matter of fact, the volume of non-rBC coatings could increase by 140% when S/C ratio increases by 20%, which will possibly lead to mass ratio of rBC to non-rBC ($M_R$ = $m_{non\text{-}BC}$ /$m_{rBC}$ ) around 2.7. Recent study (Liu et al., NG, 2017) reported that the absorption enhancement due to optical lensing effect was significantly important at $M_R$ = 3 for biomass burning aerosols. Liu et al., (2014, GRL) also reported that MCE could explain 60% of variability in single scattering albedo of biomass burning aerosols. It implies that the moderate variation in mixing state of biomass burning aerosol may have significant effect on its optical properties. As suggested by the reviewer, we will perform the optical measurement of rBC particle from biomass burning in our next research.

Liu, D., et al., (2017), Black-carbon absorption enhancement in the atmosphere determined by particle mixing state, Nature Geosci, advance online publication, 10.1038/ngeo2901

Liu, S., et al. (2014), Aerosol single scattering albedo dependence on biomass combustion efficiency: Laboratory and field studies, Geophys. Res. Lett., 41, 742–748, doi:10.1002/2013GL058392.

*Page 13, Line 14 – 15: Is this for all the data? Is there a difference in the different fuel types or MCE flaming versus smoldering conditions if the data were to be averaged from all experiments and separated into different categories? Adding the range of thicknesses sampled should also be added to the text here as the coating thicknesses look to cover the full ranges reported by the aircraft data referenced in the text.*

Reply: Yes, histogram analysis of coating thickness was performed for all the data. We found that it shows a lognormal distribution with a mode value of 20 nm and the standard deviation of 0.54. Coating thickness was mainly ranging from ~ 11 nm to 54 nm. The same analysis was also done for rBC particles with MED = 250 nm, what the difference was that histogram showed a Gaussian distribution with a mode value of ~17 nm. We will add the information in the revised manuscript.

*Page 13, Line 19 – 29: Since Figure S4 indicated coating thicknesses of 0 – at least 60 nm sampled from this data, is it possible to state that atmospheric aging results in increased rBC coatings? The data presented here and in Table 3 is from a large variety of fuel types, combustion conditions, and atmospheric aging timescales that this level of interpretation requires more investigation isolating different fuel types and atmospheric aging timescales.*

Reply: We consent to the reviewer's comment that it was hardly to attribute the difference in coating thickness of rBC particle among studies only to atmospheric aging because fuel types, combustion condition were different. We will add type of biomass in the Table 3 and revise the interpretation in the revise manuscript. Besides, we noticed that, for the rBC particles outflowed from urban area, the number fraction of thickly-coated rBC particles (S/C ratio > 2 @ $rBC_{MED}$ = 180 nm) could increased from 30% to 60% of total rBC particles as their photochemical age increased from 2h to 14 h (Moteki et al., 2007). It reflected that the photochemical process was of great importance in variation of the mixing state of rBC particles. At present, more field/airborne observational evidences of biomass burning aerosols were needed to explicit the variability in literature.

*Page 14, Line 4-5: The S/C ratio appears to increase with the EF of the NMVOC's for dry data while the wet wheat S/C does not look to depend on EF of NMVOC's. It also looks as if the S/C for the wet data is at the maximum for the dry data. More discussion of these differences should be included in the text if this is found to be significant. If not, then the wet data should be removed from the manuscript as it does not have much interpretation of the data collected here anywhere else in the manuscript.*

Reply: We will delete the data from burning of wet wheat, as suggested by the reviewer.

*Page 14, Line 9-10: Without a reference for this hypothesis or substantiation from the data presented here, this should be removed as it is too speculative.*

Reply: We will remove the sentence from the manuscript.

*Technical Corrections:*

*Page 2, Line 3: remove "the" from ". . . play a vital role in the climate change. . ."*

Reply: "the" is removed in the sentence.

*Page 2, Line 5: remove "their" from "its their"*

Reply: "their" is removed in the sentence.

*Page 2, Line 11: remove ", evidently"*

Reply: "evidently" is removed in the sentence.

*Page 2, Line 19: remove "much"*

Reply: "much" is removed in the sentence.

*Page 3, Line 27: "The variability in ΔrBC/ΔCO among observational studies also result from differences in sampling locations and conditions." – change "result" to "results"*

Reply: "result" is replaced by "results".

*Page 4, Line 10: Change ". . . we conducted open burning experiments. . . " to ". . . we conducted laboratory burning experiments. . . ".*

Reply: To avoid misleading, we replaced "open" by "laboratory" here and all other parts in the manuscript.

*Page 4, Line 20: Remove "generally"*

Reply: "generally" is removed in the sentence

*Page 5, Line 32: change "fitted" to "fit"*

Reply: "fitted" is replaced by "fit".

*Page 13,Line 9: Remove "As a matter of fact"*

Reply: "As a matter of fact" is removed in the sentence.

*Page 14, Line 5: Change "open" to "laboratory"*

Reply: "open" is changed to "laboratory" in the manuscript.

*Page 14, Line 13: Remove "urgently"*

Reply: "urgently" is removed in the sentence.

*Page 14, Line 26: Remove "obviously"*

Reply: "obviously" is removed in the sentence.

*Page 14, Line 27: Add "rBC" to say "result indicated that the rBC coagulation/growth. . ."*

Reply: "rBC" is added here.

*Table 1: Change "C/S ratio" to "S/C ratio"*

Reply: "C/S ratio" is corrected to "S/C ratio"

*Table 2: Should the last column say "MED of rBC MMD"?*

Reply: MED is abbreviation of Mass Equivalent Diameter, and MMD is Mass Mode Diameter, we will add annotation in the Table 2.

*Figure 1: Needs the information added to the figure or figure text on the different color blocks of data shown in yellow, red and blue.*

Reply: The information about the color blocks in yellow, red and blue is add in caption of Figure 1.

*Figure 2: Needs an explanation of the lines presented in the figure and the circle around one set of data. Also the text refers to 2a and 2b within the Figure which are not present.*

Reply: We will add the explanation of the lines and circle in the figure. The missing figures 2a and 2b will be added in the manuscript.

[Figure]

*Figure 4: Would changing the color scale on the number of rBC in the Figure enhance the ability for the reader to discern the two modes that can be separated with the histograms? Label the modes flaming and smoldering on the figure would also make the main points of the Figure more clear to the reader.*

Reply: We will polish the Figure 4 as suggested by the reviewers.

**Emission characteristics of refractory black carbon aerosols from fresh biomass burning: a perspective from laboratory experiments**

Xiaole Pan[1], Yugo Kanaya[2], Fumikazu Taketani[2], Takuma Miyakawa[2], Satoshi Inomata[3], Yuichi Komazaki[2], Hiroshi Tanimoto[3], Zhe Wang[1,4], Itsushi Uno[4], Zifa Wang[1]

[1]State Key Laboratory of Atmospheric Boundary Layer Physics and Atmospheric Chemistry, Institute of Atmospheric Physics, Chinese Academy of Sciences, Beijing, 100029, China

[2]Japan Agency for Marine-earth Science and Technology, Yokohama, 236-0001, Japan

[3]National Institute for Environmental Studies, Tsukuba, 305-8506, Japan

[4]Research Institute for Applied Mechanics, Kyushu University, Kasuga, 816-8580, Japan

*Correspondence to*: Xiaole PAN (panxiaole@mail.iap.ac.cn)

**Abstract.** The emission characteristics of refractory black carbon (rBC) from biomass burning are essential information for numerical simulations of regional pollution and climate effects. We conducted combustion experiments in the laboratory to investigate the emission ratio and mixing state of rBC from the burning of wheat straw and rapeseed plants, which are the main crops cultivated in the Yangtze River Delta region of China. A single particle soot photometer (SP2) was used to measure rBC-containing particles at high temporal resolution and with high accuracy. The combustion state of each burning case was indicated by the modified combustion efficiency (MCE), which is calculated using the integrated enhancement of carbon dioxide and carbon monoxide concentrations relative to their background values. The mass size distribution of the rBC particles showed a lognormal shape with an mode mass equivalent diameter (MED) of 189 nm (ranging from 152 nm to 215 nm), assuming an rBC density of 1.8 g/cm$^3$. rBC particles less than 80 nm in size (the lower detection limit of the SP2) accounted for ~5% of the total rBC mass, on average. The emission ratios, which are expressed as $\Delta$rBC/$\Delta$CO ($\Delta$ indicates the difference between the observed and background values), displayed a significant positive correlation with the MCE values and varied between 1.8 – 34 ng/m$^3$/ppbv. Multi-peak fitting analysis of the delay time ($\Delta$t, or the time of occurrence of the scattering peak minus that of the incandescence peak) distribution showed that rBC-containing particles with rBC MED = 200 ± 10 nm displayed two peaks at $\Delta$t = 1.7 μs and $\Delta$t = 3.2 μs, which could be attributed to the contributions from both flaming and smoldering combustion in each burning case. Both the $\Delta$t values and the shell/core ratios of the rBC-containing particles clearly increased as the MCE decreased from 0.98 (smoldering-dominant combustion) to 0.86 (flaming-dominant combustion), implying the great importance of the rapid condensation of semi-volatile organics. This laboratory study found that mixing state of rBC particles from biomass burning strongly depend on the its combustion processes, and

overall MCE should be took carefully into consideration while the climate effect of rBC particles from open biomass burning was simulated.

**1 Introduction**

Black carbon aerosols in the atmosphere play a vital role in climate change by absorbing solar radiation and altering the formation, lifespan and albedo of clouds (Novakov et al., 2005;Ramanathan and Carmichael, 2008;Bond et al., 2013). It was operationally defined according to its light absorption capacity, chemical reactivity and/or thermal stability (Lack et al., 2014). One definition is refractory black carbon (rBC), which corresponds to the carbon mass derived from laser-induced incandescence (LII) emission at a boiling point at 4000 K. Open biomass burning (normally refers as the burning of living or dead vegetation such as agriculture residue, grass, forest, leaves, shrub etc. due to anthropogenic activities and natural causes, abbreviated as OBB) is one of the important sources of rBC, and it contributes ~42% of atmospheric loadings in the global emissions budget (Bond et al., 2004). In addition to rBC, OBB also simultaneously emits substantial amounts of semi-volatile organics that undergo extremely complicated mixing processes with rBC during transport. Jacobson (2001) pointed out that the light absorption by internally coated rBC by inorganic/organic matter could increase, due to the "lensing effect," in which a non-absorbing coating directs more light to the cores of rBC particles. However, the debate on the absorption enhancement capacity of rBC-containing particles is still ongoing, because discrepancies exist between observations and theoretical predictions based on Mie scattering models (Shiraiwa et al., 2010;Cappa et al., 2012) and among observation results from different locations and using different sources (Healy et al., 2015;Liu et al., 2015b;Massoli et al., 2015;Ueda et al., 2016). The widely accepted explanations are as follows. First, the morphologies of rBC particles differ among different sources, and the process of particle aging in the atmosphere changes the physical structure of the particles. For instance, several studies have found that rBC particles with a fractal structure tend to collapse to a more closely packed shape when they are thickly coated (Adachi et al., 2010;Chen et al., 2010;He et al., 2015). Modeling study indicated that hygroscopic growth of rBC-containing particles also results in more compact rBC cores (Fan et al., 2016). Second, the rBC particles sometimes were also reported to be attached on the surface of non-rBC matters (Moteki et al., 2014). Either of these circumstances renders the core-shell model invalid or introduces biases into the results. Sedlacek et al., (2012) reported that a large fraction (60%) of rBC-containing particles with non-core-shell structures exist in biomass burning plumes. Methodologies have also been developed to distinguish particles with attached rBC (bare rBC on the surfaces of non-rBC particles) from rBC-containing particles with the core-shell structure (Moteki et al., 2014). In addition, OBB is an important source of brown carbon (BrC), which has distinct light absorbing features with different wavelength dependence, and their coexistence of rBC and BrC also influences the overall absorption enhancement of rBC-containing particles (Lack et al., 2012;Liu et al., 2015a;Liu et al., 2015b;Saleh et al., 2015).

[revised manuscript text omitted]

25    Table 2 summarizes the information on the sample types, the mixing ratios of CO and $CO_2$, the mass concentration of rBC particles and MCE for all burning experiments. Although the combustion proceeded generally from flaming to smoldering phase, both flaming and smoldering combustion sometimes occurred at the same time at different position of fuel. Besides, the time duration of in flaming or smoldering were different case by case. Here, the fire-integrated MCE value was used to represent the overall combustion condition for each combustion case. As shown, the averaged MCEs have no significant

30    differences for the combustion of dry wheat straw (0.86 ~ 0.98), wet wheat straw (0.88 ~ 0.96), and dry rapeseed plants (0.91 ~ 0.96). The average mass concentration of rBC after 46 times dilution ranged from 0.25 to 19.8 $\mu g/m^3$, and the average mixing ratio of CO after 22 times dilution ranged from 95 to 5003 ppbv.

**3.2 Variations in rBC size as a function of MCE**

[revised manuscript text omitted]
 5%. S/C ratios of rBC particles (Fig. 5b) were found to be 1.4 at MCE < 0.9 (smoldering combustion) and 1.2 at MCE > 0.95 (flaming combustion). Such tendency was mostly due to formation of organic matter at different combustion phase. Collier et al., (2016) reported that organic aerosol emissions had negative correlations with MCE, implying that coating processes of semi-volatile organics played a key role in the increase of S/C ratio. Variations in both the $\Delta t$ values and the S/C ratios as a function of MCE showed similar tendencies for particles with MED of the rBC cores ranging between 190 nm to 250 nm. Airborne measurements during the ARCTAS campaign (Kondo et al., 2011) showed an increasing tendency for the values of the S/C ratio ($1.3 \sim 1.66$) with increase of MCE, and the authors explained that this phenomena was because flaming phase plumes were more aged than the smoldering plumes, a 205 ± 40% increase of the volume of the coating materials resulted in a larger S/C ratio than that of rBC particles in smoldering plume.

Statistically, the modal coating thickness of rBC particles was found to be ~20 nm (Fig. S5). In fact, discrepancies exist among studies due to differences in biomass types, burning conditions and sampling locations. For example, ambient measurements in Europe indicated a coating thickness of rBC particles of 15 nm on average, and more than half of the rBC particles had a coating thinner than 10 nm (Laborde et al., 2013). Airborne measurements reported a thicker coating (65 ± 12 nm) for rBC particles from brush fires (Schwarz et al., 2008).

Table 3 summarizes recent studies that report the coating thicknesses and S/C ratios of rBC particles. Among studies, the coating thickness of freshly emitted rBC particles from burning of wheat residues was smallest with a mean value of ~ 20 nm. Coating thickness of rBC particle from brush fire (Schwarz et al., 2008) and boreal forest (Talyor et al., 2014) were 65 ±

12 nm and 50 ~ 100 nm, respectively. We noticed that there were large difference in the age of OBB plumes, and coating thickness of rBC seemed to increase as they experienced longer transport period. It implied that aging of particles was also play important role in determining the coating thickness of rBC particles. Further observational investigation on the evolution of OBB process is also need to explicate contribution to the total variability. 
[revised manuscript text omitted]

| No. | Sample* type | MCE [10th, 90th] | Duration Second | CO** ppbv | CO$_2$ ppmv | rBC*** ng/m$^3$ | rBC/CO ng/m$^3$/ppbv | rBC/CO$_2$ ng/m$^3$/ppmv | MMD**** nm | gσ |
|---|---|---|---|---|---|---|---|---|---|---|
| 1 | A | 0.964 [0.941,0.991] | 164 | 1,181 | 695 | 13983 | 24.2 | 904.8 | 215 | 1.32 |
| 2 | A | 0.930 [0.909,0.982] | 122 | 311 | 91 | 2366 | 15.6 | 1171.5 | 188 | 1.46 |
| 3 | A | 0.952 [0.884,0.973] | 94 | 261 | 114 | 1446 | 11.3 | 570.4 | 152 | 1.44 |
| 4 | A | 0.949 [0.913,0.999] | 150 | 343 | 141 | 4812 | 28.7 | 1541.0 | 187 | 1.44 |
| 7 | A | 0.953 [0.830,0.987] | 123 | 256 | 114 | 1408 | 11.2 | 554.6 | 160 | 1.42 |
| 8 | A | 0.976 [0.960,0.994] | 184 | 380 | 340 | 6290 | 33.9 | 832.6 | 191 | 1.37 |
| 9 | A | 0.917 [0.900,0.987] | 121 | 189 | 46 | 168 | 1.8 | 165.3 | 187 | 1.47 |
| 10 | A | 0.944 [0.911,0.979] | 120 | 274 | 101 | 787 | 5.9 | 348.9 | 148 | 1.43 |
| 11 | A | 0.862 [0.828,0.920] | 125 | 464 | 64 | 470 | 2.1 | 331.8 | 152 | 1.44 |
| 12 | A | 0.937 [0.853,0.988] | 151 | 230 | 75 | 429 | 3.8 | 256.5 | 148 | 1.42 |
| 13 | A | 0.950 [0.896,0.976] | 143 | 282 | 118 | 1790 | 13.0 | 682.8 | 163 | 1.46 |
| 14 | A | 0.952 [0.837,0.964] | 158 | 290 | 126 | 1295 | 9.1 | 461.1 | 160 | 1.50 |
| 15 | B | 0.909 [0.881,0.999] | 220 | 4,757 | 1,045 | 11255 | 4.8 | 484.5 | 196 | 1.33 |
| 16 | B | 0.904 [0.857,0.999] | 207 | 1,339 | 277 | 8658 | 13.2 | - | 177 | 1.36 |
| 17 | B | 0.961 [0.840,0.988] | 97 | 75 | 41 | 250 | 6.8 | 276.2 | 148 | 1.45 |
| 18 | B | 0.884 [0.730,0.999] | 230 | 1,373 | 230 | 5350 | 8.0 | 1045.7 | 181 | 1.41 |
| 19 | C | 0.943 [0.902,0.999] | 244 | 2,702 | 983 | 19802 | 15.0 | 906.0 | 204 | 1.29 |
| 20 | C | 0.923 [0.891,0.999] | 226 | 3,512 | 926 | 13402 | 7.8 | 651.2 | 189 | 1.33 |
| 21 | C | 0.909 [0.839,0.947] | 258 | 286 | 63 | 209 | 1.5 | 149.6 | 137 | 1.39 |
| 22 | C | 0.951 [0.895,0.976] | 188 | 956 | 408 | 3052 | 6.5 | 336.4 | 155 | 1.41 |
| 23 | C | 0.960 [0.874,0.985] | 172 | 457 | 241 | 1259 | 5.6 | 234.9 | 142 | 1.44 |
| 24 | C | 0.954 [0.944,0.994] | 191 | 846 | 386 | 4609 | 11.1 | 537.4 | 144 | 1.47 |

*: Sample type: Wheat straw/dry (A), Wheat straw/wet (B), Rapeseed plant/dry(C)

**: the mixing ratio of CO was diluted by 22 times.

***: the mass concentration of rBC was diluted by 46 times.

****: mode diameter in mass size distribution of rBC, abbreviated as Mass Mode Diameter

Table 3 Brief summary of the coating thickness and shell/core (S/C) ratios for rBC emissions from different sources collected from recent studies.

| rBC source | | Coating thickness | S/C ratio | rBC core size | Age | Sampling description | Study |
|---|---|---|---|---|---|---|---|
| Biomass burning | Brush fires | 65 ± 12 nm | - | *190~210 nm | 0.5~1.5 hour | Airborne SP2 measurements during 2006 Texas Air Quality Study | (Schwarz et al., 2008) |
| | | ~ 15 nm | | 200 nm | - | Field measurements using SP2 in the agglomeration of Paris as part of MEGAPOLI European project | (Laborde et al., 2013) |
| | Boreal forest | 50 ~ 100 nm | 2.0 ~ 2.5 | 152~196 nm | 1~2 days | Airborne SP2 measurement during the second phase of the BORTAS project over Eastern Canada and the North Atlantic during July-August 2011. | (Taylor et al., 2014) |
| | Agriculture | - | 1.3 ~ 1.6 | **120 ~ 140 nm | 1~2 hours | Airborne SP2 measurements during ARCTAS in spring and summer | (Kondo et al., 2011) |
| | Wheat, rapeseed plant | 20 nm (11 ~ 54 nm) | 1.2 ~ 1.4 | 200 ±10 nm | < 10 s | Burning experiments in combustion chamber in laboratory environment | This study |
| Asia continental Free troposphere | | - - | 1.6 1.3 ~ 1.4 | 200 nm 200 nm | 2~3 days 12 hours | Ground-based SP2 measurements at Fukue Island, Japan | (Shiraiwa et al., 2008) |
| Aged air mass Traffic influence | | 44 nm 2 ±10 nm | - | 200 nm 200 nm | | Field measurement using SP2 in the agglomeration of Paris as part of MEGAPOLI European project | (Laborde et al., 2013) |
| Traffic emission | | 110~300 nm | | 80~130 | Highly aged | Ground-based SP2 measurement at Urban site at Shanghai, China | (Gong et al., 2016) |
| Traffic emission Solid fuel burning Europe continental | | - - - | 1.6 ~ 2.4 <1.2 1.45 ~ 1.6 | - - - | - - - | Clean Air for London (ClearfLo) experimental campaign in winter, 2012 | (Liu et al., 2014) |
| Urban emission | | 20 ~ 30 nm | | > 200 nm | 1~2 days | Airborne SP2 measurement during MILAGRO campaign | (Subramanian et al., 2010) |

*: Volume equivalent diameter; ** Count Median Diameter

**Figures**

[Figure]

5    Figure 1: Temporal variations in the mixing ratios of CO, CO₂, the number concentrations of rBC and non-rBC particles for the burning of wheat straw (a) and rapeseed plants (b). The yellow (dry distillation step of the biomass), red (flaming-dominant combustion) and blue (smoldering-dominant combustion) shaded areas in the plot represent the different burning states.

[Figure]

Figure 2. Normalized mass size distribution (maximum value = 1) of rBC particles for each burning experiments (a). For better understanding, lognormal curve fittings are shown for a flaming-dominant combustion case (red circle) and a smoldering combustion-dominant case (blue circle). Variation of mode mass equivalent diameter (MMD) as a function of

5   the modified combustion efficiency (b). The size of circle indicates the average rBC mass concentration for each burning case. Blue dashed line and blue shaded area are the linear fitting and confidence interval for fit coefficients. The data in the gray dashed circle is excluded in the linear fitting because of their low rBC mass concentrations.

[Figure]

Figure 3. The variations in the emission ratio of rBC and ΔrBC/ΔCO, as a function of averaged MCE for all burning cases. Previous observations and the results of laboratory burning experiments are displayed in the plot.

[Figure]

Figure 4. The dependence of delay time of the peak of incandescence signal *after* that of the scattering signal as a function of the shell/core ratio for the rBC particles with MED = 200 ± 10 nm for all the burning cases, and the multiple Gaussian fitting for all the data of cross-sections along the x-axis and y-axis. For comparison with the curve fitting results, observational data

5    of flaming-dominant (MCE > 0.95, red circle) and smoldering-dominant (MCE < 0.9, blue circle) cases are also shown in the figure.

[Figure]

Figure 5. Variations in the delay time (a) and the shell/core ratio (b) as a function of MCE values for rBC particles with MED = 200 ± 10 nm.

[Figure]

Figure 6. Variations in the shell/core ratios of rBC particles with MED = 200 ±10 nm as a function of the emission factor of each experiment. Here, EF is defined as the amount of each compound released per unit amount of dry fuel consumed. The red, green and blue colors indicate the dry wheat straw, wet wheat straw and rapeseed plant samples, respectively.